 **eLIFE**

# A time-stamp mechanism may provide temporal information necessary for egocentric to allocentric spatial transformations

Avner Wallach[1,2†*], Erik Harvey-Girard[2], James Jaeyoon Jun[1‡], André Longtin[1,2,3], Len Maler[2,3]

[1]Department of Physics, University of Ottawa, Ottawa, Canada; [2]Department of Cellular and Molecular Medicine, University of Ottawa, Ottawa, Canada; [3]Center for Neural Dynamics, Mind and Brain Research Institute, University of Ottawa, Ottawa, Canada

**Abstract** Learning the spatial organization of the environment is essential for most animals' survival. This requires the animal to derive allocentric spatial information from egocentric sensory and motor experience. The neural mechanisms underlying this transformation are mostly unknown. We addressed this problem in electric fish, which can precisely navigate in complete darkness and whose brain circuitry is relatively simple. We conducted the first neural recordings in the *preglomerular complex*, the thalamic region exclusively connecting the *optic tectum* with the spatial learning circuits in the *dorsolateral pallium*. While tectal topographic information was mostly eliminated in preglomerular neurons, the time-intervals between object encounters were precisely encoded. We show that this reliable temporal information, combined with a speed signal, can permit accurate estimation of the distance between encounters, a necessary component of path-integration that enables computing allocentric spatial relations. Our results suggest that similar mechanisms are involved in sequential spatial learning in all vertebrates.
DOI: https://doi.org/10.7554/eLife.36769.001

**\*For correspondence:**
aw3057@columbia.edu

**Present address:** [†]The Zuckerman Institute, Columbia University, New York, United States; [‡]Center for Computational Mathematics, Flatiron Institute, New York, United States

**Competing interests:** The authors declare that no competing interests exist.

## Introduction

Learning to navigate within the spatial organization of different habitats is essential for animals' survival (*Geva-Sagiv et al., 2015*). Electric fish, for example, occupied a lucrative ecological niche by evolving the ability to navigate and localize food sources in complete darkness using short-range electrosensation (*Jun et al., 2016*). The spatial acuity that they exhibit, along with their reliance on learned landmark positions, strongly suggest that they memorize the relative arrangement of the landmarks and the environmental borders. The information animals use to generate such *allocentric* knowledge include sensory experiences collected during object encounters (*Jun et al., 2016*; *Petreanu et al., 2012*; *Save et al., 1998*) and motor actions (heading changes and distance traveled) executed between such encounters; utilization of these motor variables in spatial learning and navigation is termed *path integration* (*Collett and Graham, 2004*; *Etienne and Jeffery, 2004*). This acquired information, however, is always *egocentric* in nature. Fittingly, the primary brain regions dedicated to sensory and motor processing, such as the *optic tectum* (OT) of all vertebrates and many cortical regions in mammals are topographically organized along an egocentric coordinate system (*Knudsen, 1982*; *Sparks and Nelson, 1987*; *Stein, 1992*). Unravelling the neural operations underlying the transformation of egocentric sensory and motor information streams into an allocentric representation of the environment has been a central theme in studies of spatial learning and

**eLife digest** Finding their way around is an essential part of survival for many animals and helps them to locate food, mates and shelter. Animals have evolved the ability to form a 'map' or representation of their surroundings. For example, the electric fish *Apteronotus leptorhynchus*, is able to precisely learn the location of food and navigate there. It can do this in complete darkness by generating a weak electric field. As it swims, every object it encounters generates an 'electric image' that is detected on the skin and processed in the brain.

However, all the cues the fish comes across are from its own point of view – the information about its environment is processed with respect to its location. And yet, the map that it generates needs to be independent of the fish's position – it has to work regardless of where the animal is. The way animals translate 'self-centered' experiences to form a general representation of their surroundings is not yet fully understood.

Now, Wallach et al. studied how internal brain maps are generated in *A. leptorhynchus*. Information about the fish's environment passes through a structure in the brain called the preglomerular complex. Measuring the activity of this region revealed that the preglomerular complex does not process much self-centered information. Instead, whenever the fish passed any object – regardless of where it was in relation to the fish – the event triggered a brief burst of preglomerular activity. The intensity of the activity depended on how recently the fish had encountered another object. This information, combined with the dynamics of the fish's movement, could be what allows the fish to convert a sequence of encounters into a general spatial map.

These findings could help to inform research on learning and navigation. Further research could also reveal whether other species, including humans, generate their mental maps in a similar way. This may be relevant for people suffering from diseases such as Alzheimer's, in which a sense of orientation has become impaired.

DOI: https://doi.org/10.7554/eLife.36769.002

navigation. Recent studies in the mouse (*Peyrache et al., 2017*) have suggested that the key computations include vestibular input that defines the animal's head direction (head direction cells, egocentric) and external sensory input that signals the presence of stable environmental features (i.e., landmarks). Linking the head directions that orient the animal to different environmental features are then hypothesized to generate an allocentric representation of those features. The neural circuits that have been hypothesized to implement these computations are, however, exceedingly complicated and include thalamic (head direction) and cortical (external sensory) input to the hippocampal formation. The proposed wiring diagrams are highly speculative and very far from providing a well-defined mechanistic model of how spatial maps are created.

Equivalent computations appear to be carried out in Drosophila (*Seelig and Jayaraman, 2015*). Visual orientation to landmarks and body direction via path integration are combined in the ellipsoid body with dynamics suggestive of a ring attractor. While these studies in the simpler nervous system of the fly are now closer to providing a mechanistic explanation of how egocentric and external (visual) inputs are combined, it is not clear if the fly has a full representation of the allocentric relations of different environmental features. It is also not at all clear that the dynamics of the ellipsoid body can be mapped onto the cortical and hippocampal circuitry of mammals.

Teleost fish offer an attractive model for studying this question, as their related brain circuitry is relatively tractable: lesion studies point to the dorsolateral pallium (DL) as the key telencephalic region required for allocentric spatial learning (*Broglio et al., 2010*; *Durán et al., 2010*; *Rodríguez et al., 2002*), similarly to the medial cortex in reptiles and the hippocampus in mammals (see Discussion). DL has strong excitatory recurrent connectivity (*Elliott et al., 2017*; *Giassi et al., 2012b*). Importantly, DL receives sensory and motor information related to electrosensory and visual object motion from OT (*Bastian, 1982*) via a single structure – the diencephalic preglomerular complex (PG, *Giassi et al., 2012b*, *Figure 1A*). The tectal recipient portion of PG projects solely to DL (*Giassi et al., 2012a*) in agreement with the lesion studies. Importantly, PG receives very little feedback from areas associated with DL (*Giassi et al., 2012b*) and therefore functions as an exclusive feed-forward bottleneck between OT and the pallium. DL in turn projects to the central pallium

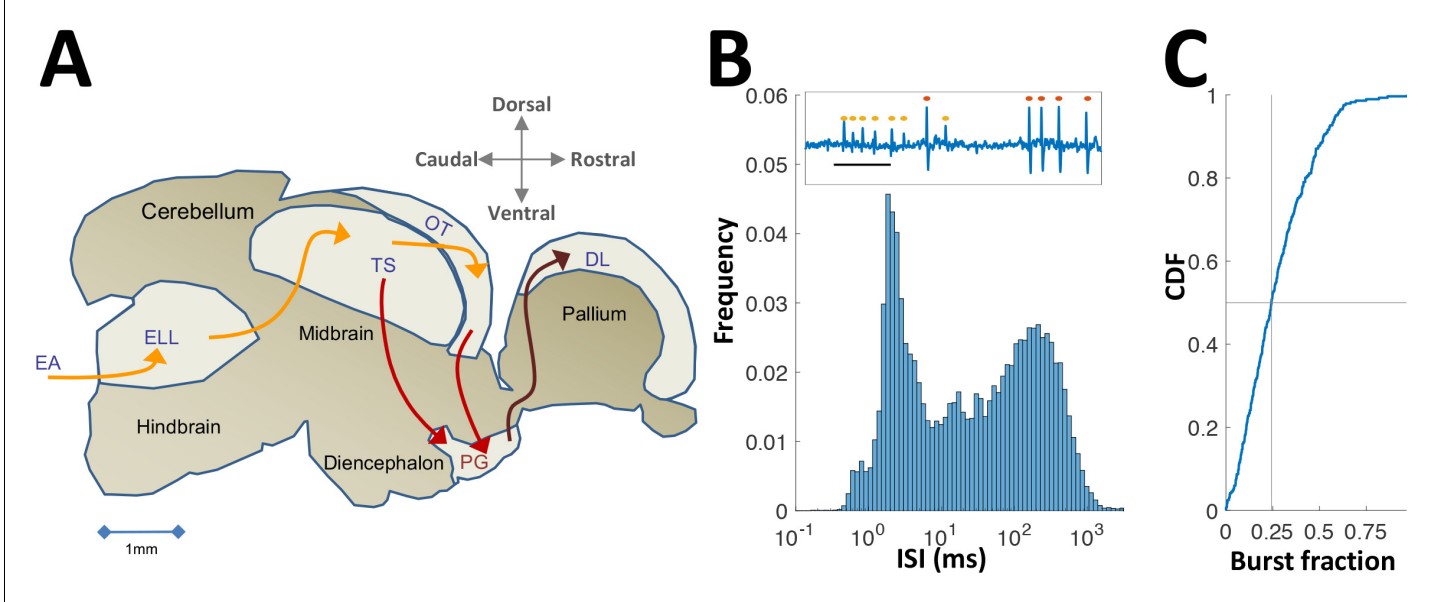

**Figure 1.** Neural recordings in PG. (**A**) Electrosensory pathways from periphery to telencephalon. EA, Electrosensory afferents; ELL, electrosensory lobe; TS, Torus semicircularis (similar to the inferior colliculus); OT, optic tectum (homolog of the superior colliculus); PG, preglomerular complex; DL, dorsolateral pallium. (**B**) Interspike-interval (ISI) distribution in an example PG cell. Note peak around 2 ms due to thalamic-like bursting. Inset: extracellular voltage example depicting two units (red and yellow markers); scale-bar, 5 ms. (**C**) Cumulative distribution function of burst fraction (see Materials and methods) for all PG single-units; median = 0.24.

DOI: https://doi.org/10.7554/eLife.36769.003

The following figure supplements are available for figure 1:

**Figure supplement 1.** T-type $Ca^{2+}$ channel expression profiles support PG thalamic homology.
DOI: https://doi.org/10.7554/eLife.36769.004
**Figure supplement 2.** Dataset summary.
DOI: https://doi.org/10.7554/eLife.36769.005
**Figure supplement 3.** Spike sorting example.
DOI: https://doi.org/10.7554/eLife.36769.006
**Figure supplement 4.** Localization of recorded units.
DOI: https://doi.org/10.7554/eLife.36769.007

(DC, *Giassi et al., 2012b*); DL also has reciprocal connections with the dorsal pallium (DD) and DD itself has strong recurrent connectivity. DC is the only route by which DL can control motor activity and it does so solely via its projections to the OT (*Giassi et al., 2012b*). We hypothesize that ego-centric object-related information (OT) conveyed by PG to DL, is converted to a learned allocentric spatial map by the recurrent circuitry of DL, DD and DC; DC, in turn, then controls the fish's spatial behavior via its projections to OT.

## Results

In what follows we describe the first electrophysiological recordings conducted in PG of any fish species. PG has been considered part of the fish thalamus, based on simple anatomical criteria (*Giassi et al., 2012a*; *Ishikawa et al., 2007*; *Mueller, 2012*). Our recordings provided an additional functional correspondence in that PG cells emit rapid spike bursts likely mediated by the T-type $Ca^{2+}$ channels (*Figure 1B,C* and *Figure 1—figure supplement 1*) characteristic of the OT targets in thalamic regions of other vertebrates (*Ramcharan et al., 2005*; *Reches and Gutfreund, 2009*). In this contribution, we show a radical conversion of the topographic spatial representation in OT into a reliable non-topographic temporal representation of encounter sequences. We then use a computational model to demonstrate that this temporal information is readily accessible for decoding and that it is sufficiently accurate to account for spatial precision during naturalistic behavior (*Jun et al., 2016*).

We characterized the responses of 84 electrosensory PG cells of the weakly electric fish *Apteronotus leptorhynchus* to moving objects; several motion protocols were used and the sample size for each protocol is mentioned in context (*Figure 1—figure supplement 2*). Neuronal activity was recorded extracellularly (*Figure 1—figure supplement 3*) in immobilized fish while objects (brass or plastic spheres) were moved relative to the skin using a linear motor (Methods). Cells responding to this stimulation were predominantly found in the lateral nucleus of the PG complex, PGl (*Figure 1—figure supplement 4*).

## Topographic spatial information is abolished in PG

We first examined the spatial representation in PG cells by measuring their receptive fields (RFs; measured in 27 cells). The OT electrosensory cells driving PG have spatially-restricted, topographically organized RFs (*Bastian, 1982*), and thus provide labeled-line information on the egocentric position of objects. In PG, by contrast, only 11% of the cells (3/27) were topographic with a spatially restricted RF (*Figure 2A*); the majority of PG cells (89%, 24/27) responded across most or all of the fish's body (*Figure 2B* and *Figure 2—figure supplement 1*). Therefore, PG activity does not convey a topographic 'labeled-line' code of object position. We also checked whether object location is encoded by the firing-rate of PG neurons (i.e., a rate code). The firing-rate was significantly correlated with object position only in one non-topographic cell (4%, 1/24 neurons; $p < 0.05$, random-permutations test, *Figure 2C*). Similarly, mutual information between object position and firing-rate was significant only in two non-topographic cells (8%, 2/24 neurons; $p < 0.05$, random-permutations test, *Figure 2D*). Therefore, almost all PG cells have whole-body RFs and lack topographic spatial information– the hallmark of all electrosensory regions from the sensory periphery up to OT. We attempted to systematically sample throughout the full extent of PG. However, we cannot absolutely determine whether the subset of topographic cells represent a small distinct sub-nucleus or are sparsely distributed throughout the PG complex.

## PG cells respond to object encounters

Next, we checked what information PG neurons convey about object motion. OT cells, which drive PG, respond to an object moved parallel to the fish (longitudinal motion) while the object traverses their RFs (*Bastian, 1982*). Only a minority of recorded PG units (26%, 7 out of 27 tested with longitudinal motion) responded in this manner (*Figure 3A*). Rather, the majority of PG units (78%, 20/27) exhibited a strikingly different behavior, emitting a brief burst response confined to the onset (and sometimes to the offset) of object motion, but not during motion itself (*Figure 3B*). This is further demonstrated when motion in each direction was broken into four segments separated by wait periods (*Figure 3C*), evoking responses at the onset (yellow arrowheads) and offset (red arrows) of each segment across the entire body. Interestingly, we have encountered several lateral-line responsive PG units that, unlike most of their electrosensory counterparts, did respond persistently throughout (and even after) object motion (*Figure 3—figure supplement 1*).

We next applied transverse motion, which mimics an object looming/receding (incoming/outgoing) into/from the electrosensory receptive field (tested in 40 cells). Three types of responses to such motion were identified: proximity detection, encounter detection, and motion-change detection; 50% (20/40) displayed more than one type of response. Proximity detectors (75%, 30/40) responded when an object was encountered very close to the skin (< 2 cm, *Figure 3D*); encounter detectors (32%, 13/40) responded when an object either entered to or departed from their electroreceptive range (~4 cm, *Figure 3E*); lastly, motion-change detectors (57.5%, 32/40) displayed a response similar to that observed in longitudinal motion, firing at the onset/offset of motion (i.e., when the object accelerated/decelerated, *Figure 3F*); remarkably, this type of response was relatively distance-invariant, yielding comparable responses both very close to (0.5 cm) and very far from (5.5 cm) the skin despite the drastic effects of distance on both the magnitude and spread of the object's electrical image (*Figure 2—figure supplement 1*, see *Chen et al., 2005*). This discrete, nearly stereotypic response stands in contrast to the finely-tuned rate code of object distance previously reported in the primary electrosensory cells in ELL (*Clarke et al., 2014*; *Clarke and Maler, 2017*). Taken together, the variety of PG response-types mentioned above may constitute a distributed representation of object proximity. Unlike the loss of egocentric topographic mapping discussed in the

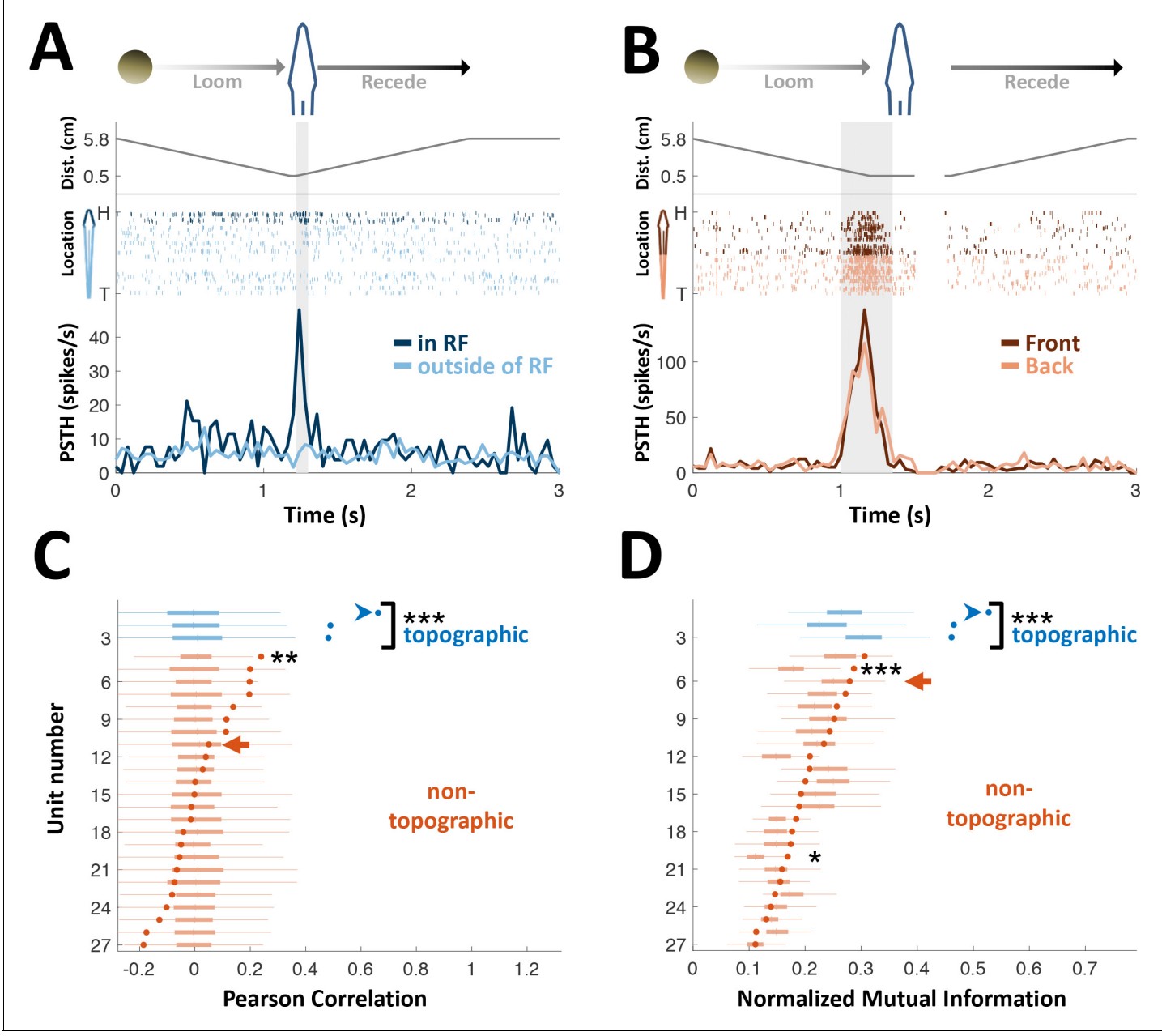

**Figure 2.** PG cells lack egocentric spatial information. (A) Example of an 'atypical' topographic PG cell with a restricted receptive field (RF) at the head. Looming-receding motions were performed at different locations. Top: object distance from skin vs. time; middle: spike raster plot, arranged according to object location (H = head, T = tail); bottom: PSTH within the RF (dark blue) and outside of it (light blue). (B) Example of a 'typical' non-topographic PG cell, responding to stimulation across the entire body. Bottom: PSTH of front (dark red) and rear (light red) halves of fish. RF extended beyond the sampled range. (C) Circles: Pearson correlation between normalized egocentric location and firing-rate for all units, in ascending order; box-plots: distribution of correlations obtained by random permutations of locations (control). Only one non-topographic cell had correlations that statistically differed significantly from those in the randomized boxes (**). (D) Circles: normalized mutual information (MI) between egocentric location and firing-rate for all units, in ascending order; box plots, distribution of MI obtained by random permutations of locations (control). Only two non-topographic cells had MI that statistically differed significantly from those in the randomized boxes (* and ***). Blue arrowheads in (C) and (D) mark the cell depicted in (A); Red arrows mark the cell depicted in (B). P values obtained using permutation test: '*', p < 0.05; '**', p < 0.01; '***', p < 0.001.

DOI: https://doi.org/10.7554/eLife.36769.008

The following figure supplement is available for figure 2:

**Figure supplement 1.** Receptive fields across the PG population.

DOI: https://doi.org/10.7554/eLife.36769.009

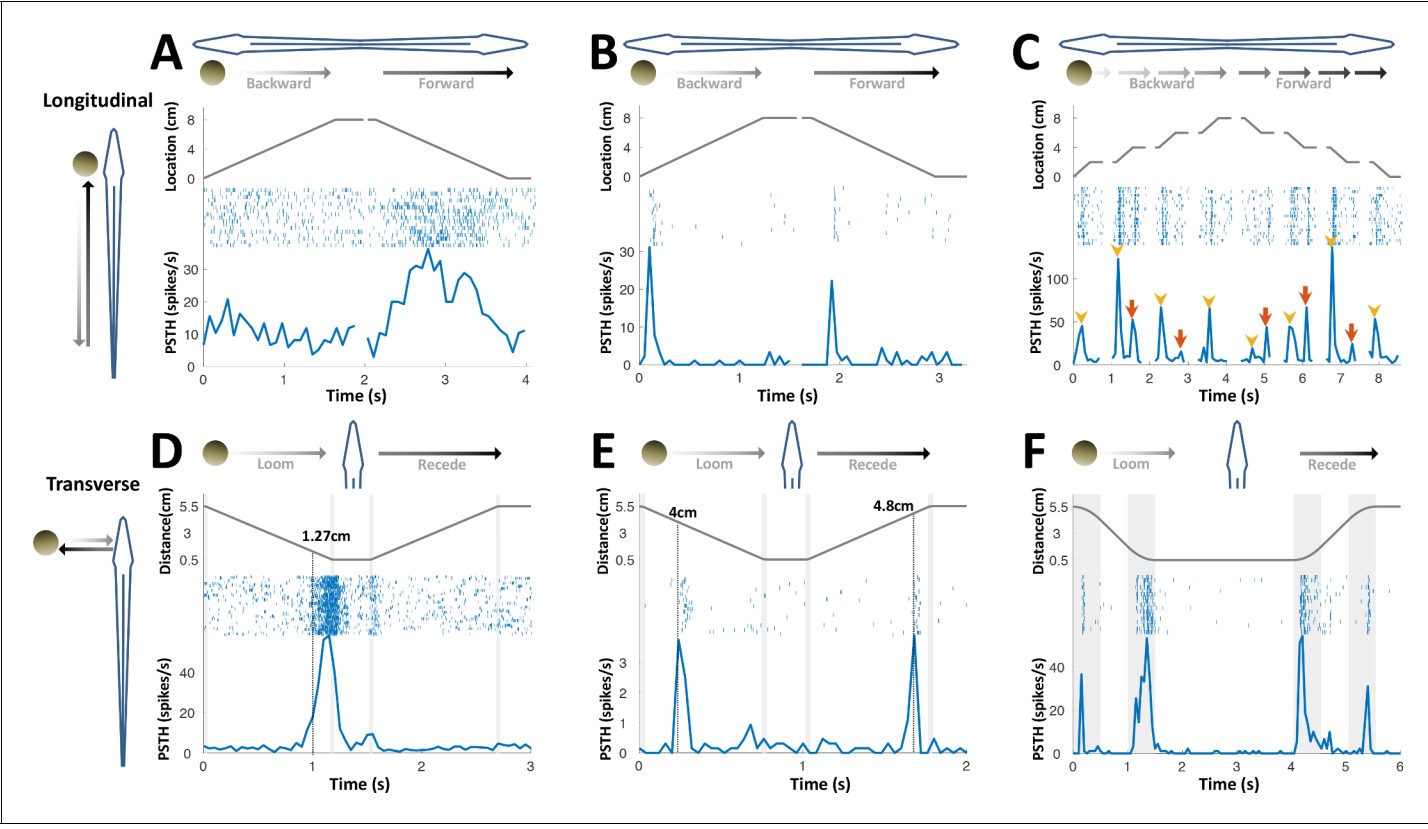

**Figure 3.** PG cells respond to motion novelty. (A–C) Longitudinal motion. Top: object location (distance from the tip of the nose); middle: spike raster; bottom: PSTH. (A) 'Atypical' PG cell responding uni-directionally, throughout the motion toward the head. 43% (3/7) of atypical PG cells responded during forward motion only, while 57% (4/7) responded in both directions. (B) 'Typical' PG cell responding to motion-onset at the head during backward motion and to motion-onset at the tail during forward motion. 75% (15/20) of these motion novelty cells respond bi-directionally, 10% (2/20) only during forward motion and 15% (3/20) only during backward motion. (C) PG cell responding to motion onset (yellow arrowheads) and to motion offset (red arrows) in intermittent-motion protocol. (D–F) Responses to transverse (looming-receding) motion. All PG cells recorded with transverse-motion (40/40) responded to object looming, while only 40% (16/40) responded to receding. Top: object distance from skin; middle: spike raster; bottom: PSTH. Grey shadings mark acceleration/deceleration. (D) Proximity detector cell. Note that the response starts prior to the deceleration (shaded). (E) Encounter detector cell. (F) Motion change detector cell, responding when objects accelerated/decelerated (shaded, 10 cm/s$^2$ in this example) similarly to the response observed to longitudinal motion (panels B and C).

DOI: https://doi.org/10.7554/eLife.36769.010

The following figure supplements are available for figure 3:

**Figure supplement 1.** PG lateral-line responses.
DOI: https://doi.org/10.7554/eLife.36769.011

**Figure supplement 2.** The amplitude (left ordinate, log scale) and spread (right ordinate, linear scale) of the object's electric image on the skin, as functions of distance, based on a previously published model of electric fields in this fish (*Chen et al., 2005*).
DOI: https://doi.org/10.7554/eLife.36769.012

previous section, this representation does provide DL with coarse-grained, non-directional egocentric information. The possible role(s) of this information was not further considered in our analyses.

We conclude that PG electrosensory cells predominantly respond to novelty: onset/offset of object motion within their receptive field or the introduction/removal of objects into/from this receptive field; we use the term object encounters to designate all such events. Most PG cells responded both to conductive and non-conductive objects (*Figure 4*) and would therefore not discriminate between different object types, for example, plants versus rocks. We can, therefore, infer that during active exploration of the environment, PG reports to the dorsal pallium whenever the fish encounters or leaves a prey or a landmark (e.g., root mass or rock) or alters its swimming trajectory near a landmark.

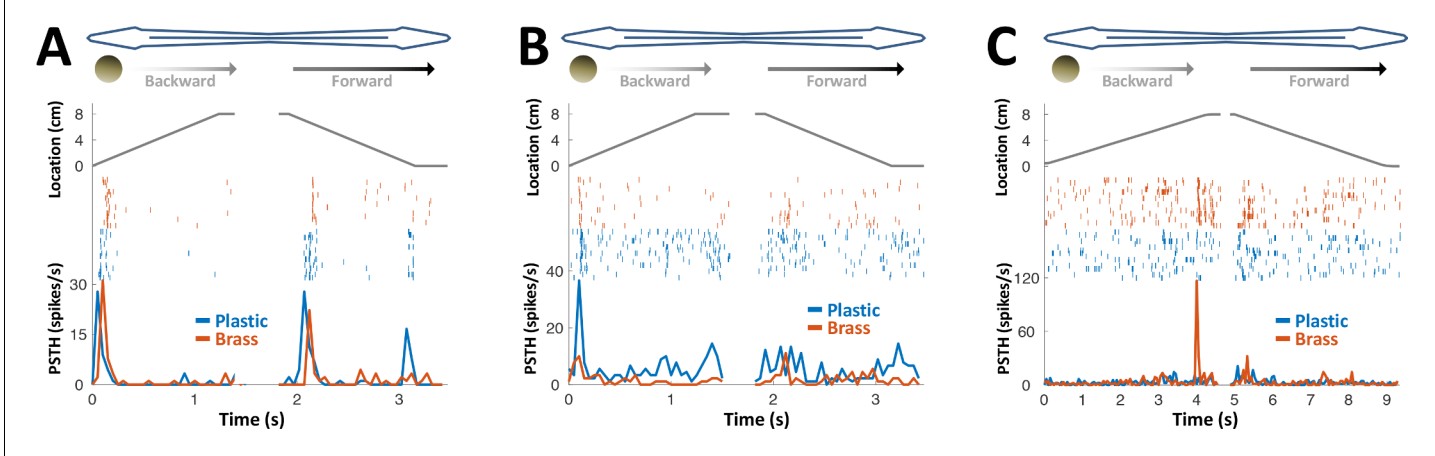

**Figure 4.** PG response to conductive/non-conductive objects. (A–C) Top: object location (distance from the tip of the nose) during longitudinal object motion; middle: spike raster; bottom: PSTH. Red: conductive object (brass sphere, mimicking plants and animals); Blue: non-conductive object (plastic sphere, mimicking rocks). (A) Example cell responding to both brass and plastic spheres (such responses were found in 57% of the cells, 24/42). (B) Example cell responding primarily to plastic, with a very faint response to brass (5% of the cells, 2/42). (C) Example cell responding to brass only (38% of the cells, 16/42). The predominance of cells responding to both brass and plastic spheres or to brass-only spheres is likely inherited from OT (*Bastian, 1982*).

DOI: https://doi.org/10.7554/eLife.36769.013

## PG cells display history-independent, spatially non-specific adaptation

While topographic egocentric information was scarce in PG, temporal information was prevalent. Many of the PG cells exhibited pronounced adaptation to repeated motion (45%, 15/33 neurons tested with repeated-motion protocol; *Figure 5—figure supplement 1*). We delivered a sequence of object encounters (motions in and out of the RF, *Figure 5A*); time-intervals between sequential encounters were drawn randomly and independently (in the range of 1–30 s). The response intensity (number of spikes within an individually determined time window) of these adapting cells to each encounter was strongly correlated with the time-interval immediately prior to that encounter (*Figure 5B*). By contrast, there was no significant correlation with intervals prior to the last one (inset of *Figure 5B* and *Figure 5C*). Thus, a large subset of PG cells encoded the duration of the time-interval *preceding the last encounter* – but did not convey information about the intervals prior to that. We term this history-independent adaptation. Interestingly, similar behavior was also found in visually-responsive cells (ventral PG, *Figure 5—figure supplement 2*). This result contrasts with many studies in the mammalian cortex, where adaptation is incremental and responses depend on multiple preceding intervals (*Lampl and Katz, 2017*; *Ulanovsky et al., 2004*). Moreover, a spatial 'oddball' experiment (Methods), which we conducted on 14 neurons (6 of which were adapting) revealed that this adaptation is also spatially non-specific – that is, encounters at one body location adapt the response to encounters at all other locations (*Figure 3D–F*). Therefore, this thalamic adaptation mechanism enables neurons to encode the time between the two most recent successive stimuli, without contamination by prior encounter history and irrespective of the objects' egocentric position.

## PG cells convey readily accessible temporal information

Can the sequence of time intervals between encounters be accurately decoded from the activity of the adapting PG cells? To answer this question, we first constructed a simple model to describe the cells' response dynamics. This model assumes each cell's responsiveness is governed by a resource variable $x$ which is depleted following each encounter and recovers exponentially between encounters (*Tsodyks and Markram, 1997*). A parameter $\beta$ indicates the fraction of the resource remaining after depletion, while a time-constant $\tau$ sets the speed of recovery. The resource variable $x$ is converted into a firing rate variable $\lambda$ using static rectification with a gain parameter $a$ and baseline activity parameter c. Finally, each encounter generates a spike count according to a Poisson

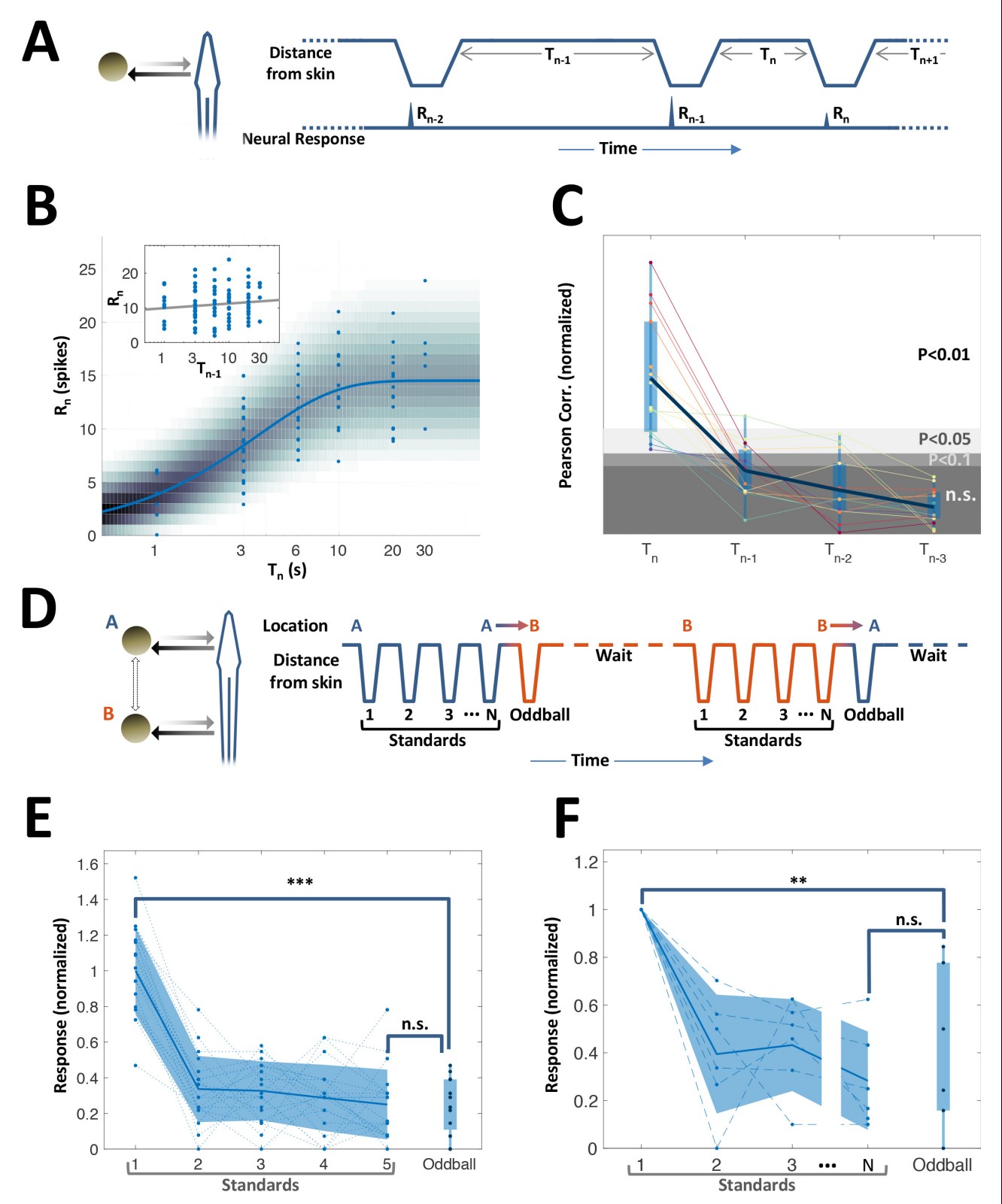

**Figure 5.** History-independent, spatially non-specific adaptation of PG cells. (**A**) Random interval protocol. Object motions toward- and away-from-skin generated responses $\{R_n\}$, measured by spike count within individually determined time-window. Motions were interleaved by random time intervals $\{T_n\}$ (1–30 s). (**B**) Response versus the preceding time-interval (log scale) for an example neuron (circles). Model fitted to the data-points is depicted by response rate (solid blue line) and spike count distribution (background shading); correlation of data to model = 0.56 (p < 10$^{-6}$, random permutations).
*Figure 5 continued on next page*

*Figure 5 continued*
Inset: same data-points plotted as function of penultimate intervals; grey line: linear regression to log(T) (correlation = 0.15, p = 0.136, random permutations). (C) Correlations between responses and time-intervals (last interval and one-, two- or three-before-last), normalized to the 10, 5 and 1 top percentiles of correlations generated by random permutations of the data, for each of the 15 adapting cells tested (thin colored lines; thick blue line: mean across population). For most cells, the response was significantly correlated only with the last interval. (D) Spatial oddball protocol. A sequence of N standard stimuli at one location (N = 5–9) was followed by one oddball stimulus at a different location. (E) Response of example cell (normalized to non-adapted response) to oddball protocol (dotted lines: individual tirals, N = 10; thick line and shaded area: mean ± standard deviation). Response to the oddball was significantly weaker than the non-adapted response (p < 10$^{-4}$, bootstrap) and not significantly different from the response to the last standard stimulus (p = 0.45, bootstrap). (F) Normalized response of all adapting cells tested with the spatial oddball protocol (dashed lines: individual cells, N = 6; thick line and shaded area: mean ± standard deviation). Population response to the oddball was significantly weaker than the non-adapted response (p = 0.001, bootstrap) and not significantly different from response to the last standard stimulus (p = 0.2, bootstrap). '**', p < 0.01; '***', p < 0.001; n.s., not significant.
DOI: https://doi.org/10.7554/eLife.36769.014
The following figure supplements are available for figure 5:
**Figure supplement 1.** Response adaptation to periodic motion.
DOI: https://doi.org/10.7554/eLife.36769.015
**Figure supplement 2.** PG Visual responses.
DOI: https://doi.org/10.7554/eLife.36769.016

distribution with the current value of the rate parameter $\lambda$ (*Figure 6A*). The parameter $\beta$ determines the extent to which past intervals in the sequence affect the current state (*Figure 6B*). For instance, $\beta=0$ signifies complete exhaustion of the resource at each and every encounter. For such a cell, therefore, the firing rate increases monotonically as a function of the last time interval (with the dependence taking the form of a saturating exponential function) regardless of past intervals. In other words, $\beta = 0$ indicates a completely history-independent cell, as defined above. As $\beta$ approaches unity the memory of past intervals becomes more dominant, for example, a succession of short intervals will yield progressively weaker responses.

All 15 adapting cells had statistically significant correlations between the responses and the fitted rate parameter $\{T_n\}$ (p < 0.05, random permutations, see *Figure 6—figure supplement 1A, B*). Consistent with the results in the previous section, most adapting cells had an extremely low value of $\beta$ (see parameter distributions in *Figure 6—figure supplement 1*); 33% of the cells (5/15) were best fitted with $\beta$ = 0 (see example in *Figure 5B*), the median $\beta$ was 0.12 and only 13% (2/15) had $\beta > 0.5$. Hence, most cells were indeed predominantly affected only by the very last time interval. The fitted recovery time-scales were in the range 2.6-25.3s (see *Figure 6—figure supplement 1C*). Note, however, that this largely reflects the range explored with our stimulation protocol (1-30s intervals); it is quite probable that PG contains faster cells (i.e., with $\tau < 1s$) that appear to be 'non-adapting' under this protocol, as well as slower cells (i.e., with $\tau > 30s$) that quickly cease responding and were therefore not recorded from.

Using this model, we simulated the response of a population of adapting PG cells to random interval sequences. A Maximum-Likelihood Estimator (MLE, see Materials and methods) was used to decode the most recent interval from the population response to each encounter. To demonstrate this approach, we used a homogeneous population of identical memoryless neurons ($\beta = 0$). At intervals shorter than the cells' adaptation time-constant $\tau$ the MLE was approximately unbiased (*Figure 6C*) and saturated the Cramér-Rao lower error bound (CRLB, *Figure 6D*). The error increased with the decoded time interval $T$ and baseline spontaneous activity $c$ and decreased with population size $N$ and response gain $a$ (*Figure 6E-G*). Finally, we checked the effect of non-zero adaptation memory ($\beta > 0$) while still using the same memoryless MLE decoder (constructed by fitting a memoryless model to the simulated responses). The estimation error and bias were nearly unchanged up to $\beta = 0.2$ (*Figure 6H*). We conclude that the response of the majority of PG adapting cells to an encounter can be used to estimate the time interval elapsed prior to that encounter, using a simple, memoryless decoder.

It is important to note that the population model mentioned above assumes that the responses of individual cells are statistically independent given the interval sequence. This assumption may not hold in certain scenarios, such as the existence of a latent state variable (e.g. arousal) modulating the overall population responsiveness. Such correlations across the population, if not taken into

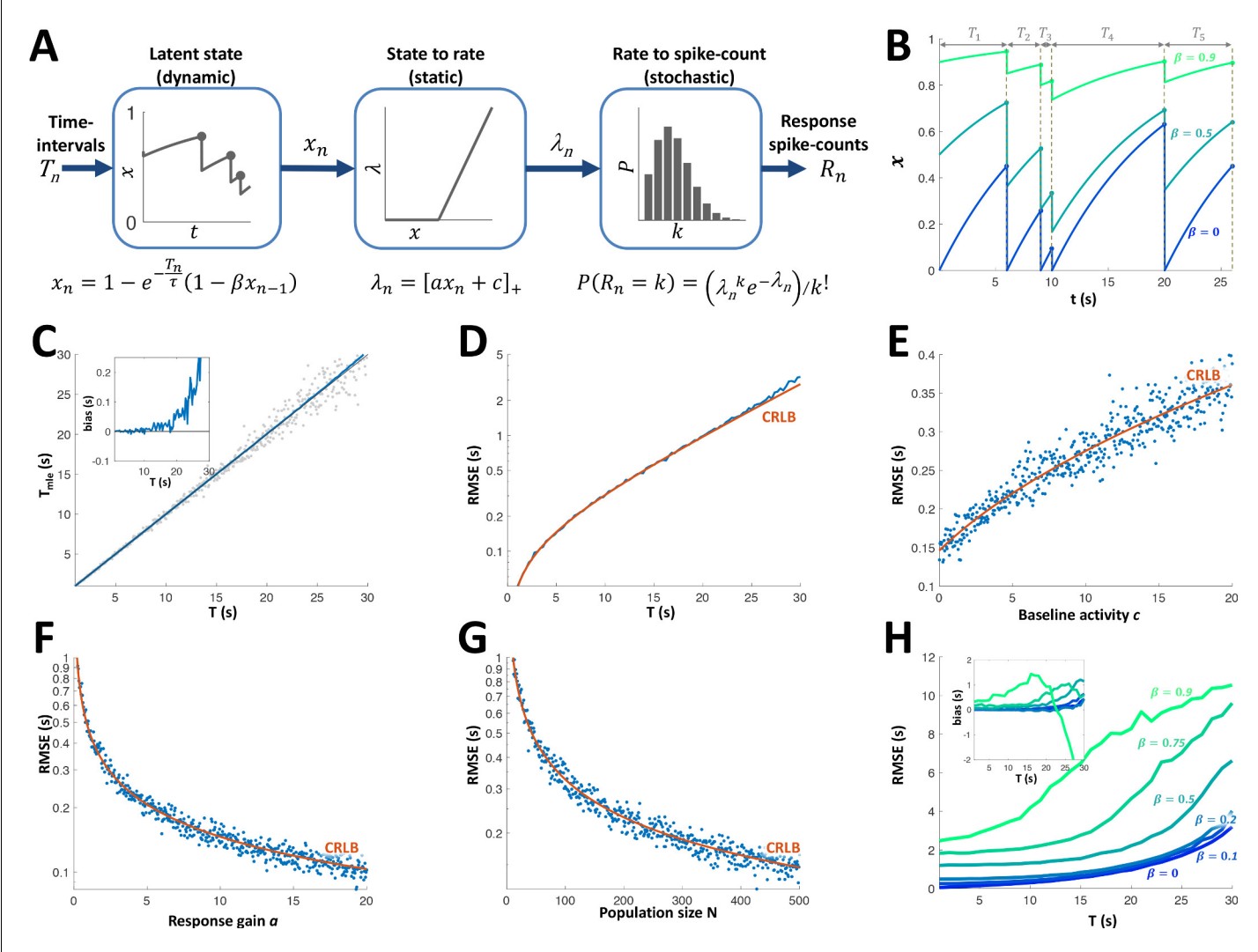

**Figure 6.** Adaptation model and Maximum Likelihood Estimation of time inteval (**A**) Block diagram of adaptation model, which includes three stages: a dynamic latent state variable $x$ with memory parameter $\beta$ and exponential recovery with time-constant $\tau$; a static non-negative mapping from state $x$ to firing parameter $\lambda$, with gain parameter $a$ and baseline parameter $c$; and a stochastic spike-count generator (Poisson random variable with parameter $\lambda$). (**B**) Examples of state variable dynamics due to a random encounter sequence, for different values of $\beta$. Note that when $\beta = 0$, each encounter entails complete reset of $x$ to 0, thus generating the history independence property. (**C**) MLE of time interval with a homogeneous population. Grey circles: individual estimations; blue curve: mean; black line: identity diagonal. Inset: estimation bias. (**D**) Blue: root-mean-squared estimation error (RMSE, log scale) vs. time interval. Red: Cramér-Rao Lower Bound (CRLB). (**E**) RMSE vs baseline activity $c$. (**F**) RMSE (log scale) vs. response gain $a$. (**G**) RMSE(log scale) vs. population size N. (**H**) RMSE for different values of $\beta$. Inset: estimation bias. In all simulations $\beta = 0$, $\tau = 10s$, $a = 10$, $c = 0$, $N = 500$, $T = 5$ unless stated otherwise.

DOI: https://doi.org/10.7554/eLife.36769.017

The following figure supplement is available for figure 6:

**Figure supplement 1.** Empirical distributions of model parameters.

DOI: https://doi.org/10.7554/eLife.36769.018

account in the decoding algorithm, would degrade the estimation accuracy. One possible solution to this could be based on the population of *non-adapting cells* (55%, 18/33) which, by definition, are not significantly affected by the temporal pattern of stimulation but might be affected by such global modulations. Thus, this population may convey sufficient information to enable estimation of such confounding variables and correct the temporal estimation accordingly.

## PG activity corresponds to observed path integration acuity

Can PG activity be used to explain features of spatial behavior in freely-swimming electric fish? To check this possibility, we must first measure the time-intervals being memorized by fish conducting a spatial learning task, as well as the error in the fish's estimation of these intervals. To this end we used previously published data from spatial learning experiments conducted in our lab (*Jun et al., 2016*). In these experiments fish were trained to find food in a specific location in complete darkness. This was performed either with or without the aid of landmarks. Since only the short-range electrosense was available to the fish, they had to use path-integration from the encountered objects to the food location. In other words, the fish had to memorize the intervals/distances between these encounters in order to improve its behavioral performance and find the food more efficiently. Therefore, we measured the time-interval and the distance traveled by the trained fish from the last encounter with an object (tank wall or landmark) to the food in each trial, either with (*Figure 7A*) or

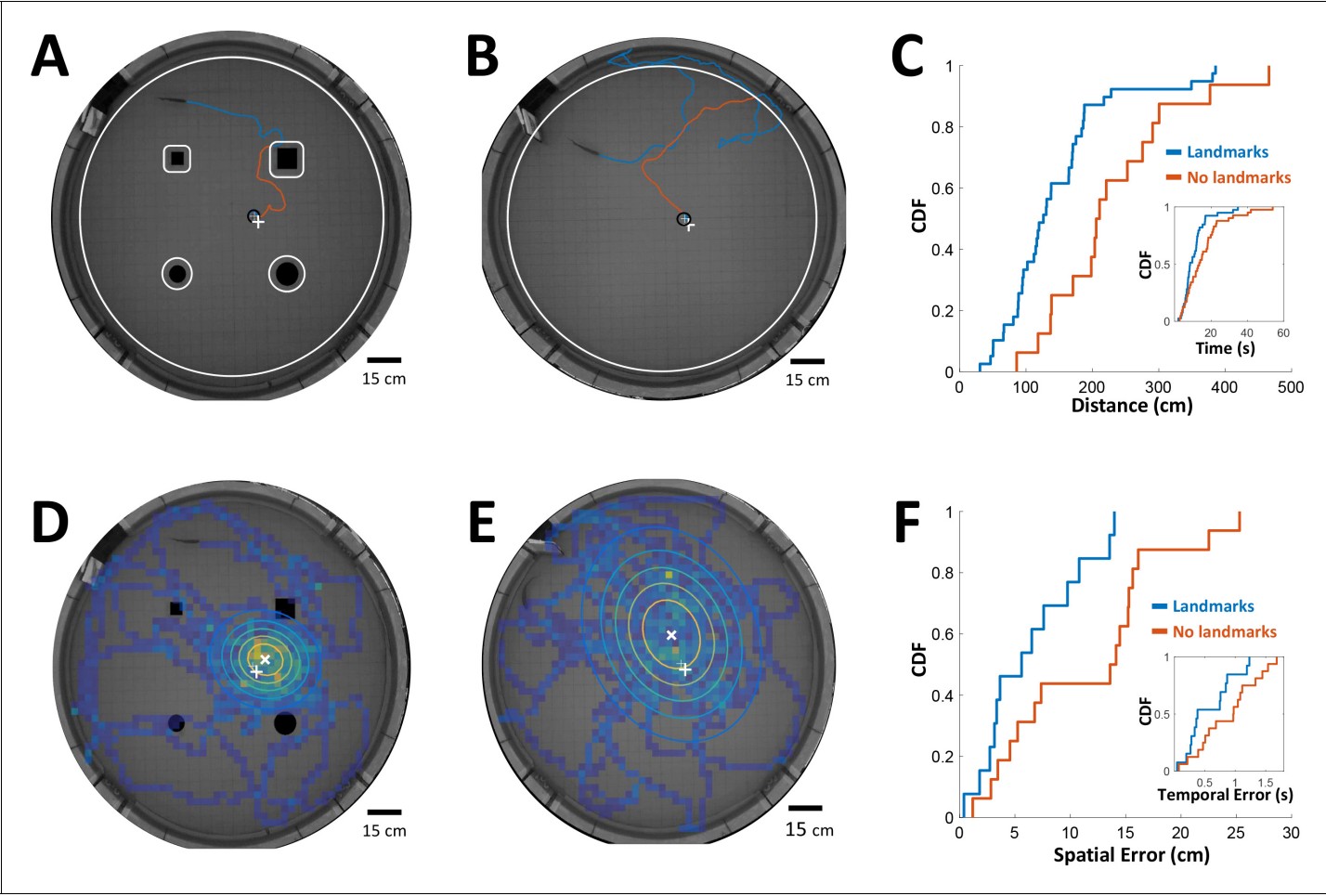

**Figure 7.** Path integration accuracy measured in behaving fish. (**A**) Fish navigating to food placed at designated location (white '+' sign) with the aid of landmarks (black silhouettes). Orange line: cruising distance from the last encounter (3 cm from landmark or 6 cm from wall, white contours) to the detection of the food (3 cm from food, black contour around white '+' sign). (**B**) Same as A in an experiment with no landmarks. Note the increase in traveled path. (**C**) Cumulative distribution functions (CDF) of the distances from last encounter to food, with (blue) and without (red) landmarks. Inset: CDFs of time elapsed from last encounter to food. (**D**) Fish searching for missing food in a probe trial with landmarks. Heat map: visit histogram; colored ovals: 2-D Gaussian fitting of histogram; the spatial error is the distance from fitted Gaussian center (white 'x') to the memorized food location (white '+'). (**E**) Probe trial without landmarks. Note the increase in error compared with panel D. (**F**) Cumulative distribution functions (CDF) of the fish's spatial errors, with (blue) and without (red) landmarks. Inset: CDFs of the temporal errors, obtained by dividing the spatial errors by the fish's average velocity.

DOI: https://doi.org/10.7554/eLife.36769.019

without landmarks (***Figure 7B***). As one might expect, these intervals were longer in the absence of landmarks (***Figure 7C***).

Next, in order to estimate the fish's temporal accuracy, we randomly performed 'probe' trials in which no food was present in the arena. In these trials the fish vigorously searched for the missing food around its estimated position. We measured the distance between the center of the searched area and the designated food location to obtain the fish's *spatial error* (***Figure 7D***, with landmarks; ***Figure 7E***, without landmarks). This error increased in the absence of landmarks as well (***Figure 7F***). Dividing this error by the median velocity in each trial yielded the *temporal error*, that is the error in the fish's estimation of the time elapsed between object encounters and the estimated food location (***Figure 7F***, inset). Thus, this behavioral analysis yielded the relation between the memorized time-intervals and the estimation error of this variable in two different scenarios.

PG is the only source of sensory information for the dorsolateral pallium (DL), the likely site of spatial memory (Rodriguez et al., 2002). Therefore, we hypothesized that the history-independent adapting PG cells provide DL with one *necessary* component for path integration: the temporal sequence of object encounters (the two other components are heading-direction and linear velocity). How many adapting PG cells are required to achieve the behavioral temporal acuity observed as described above? We constructed a heterogeneous population model using the empirically obtained parameters ($\beta$ was set to 0 for all neurons to simplify the simulations). This model was then stimulated at random time-intervals, which were then decoded from the population response using MLE. This estimation was approximately unbiased across the entire range tested (***Figure 8A***). We compared the simulated estimation error, as well as the analytically computed CRLB for the heterogeneous population, to the behavioral temporal results obtained with and without landmarks (***Figure 8B***). We also performed the comparison in spatial terms by using the fishes' average velocity to convert the model's results into units of distance (***Figure 8C***). The simulated PG population yielded temporal and spatial estimation errors comparable to those displayed in behavior (boxplots in ***Figure 8B, C***), both with and without landmarks, using only ~500 adapting cells. The lateral subdivision of PG (PGl), in which most motion-responsive cells were found, contains about 60,000 cells (***Trinh et al., 2016***). Based on our data, we can estimate that over 9,000 of these are memoryless adapting cells (a conservative estimate using only strictly history-independent cells with $\beta = 0$, 5/33 or 15%). Thus, we hypothesize that the number of PGl cells is *sufficient* to attain the observed behavioral precision, even when additional encoding errors (e.g. in heading and velocity estimation) are taken into account. Taken together, our simulations suggest that time-interval encoding in PG can consistently account for the observed behavioral precision of spatial learning. It should be noted

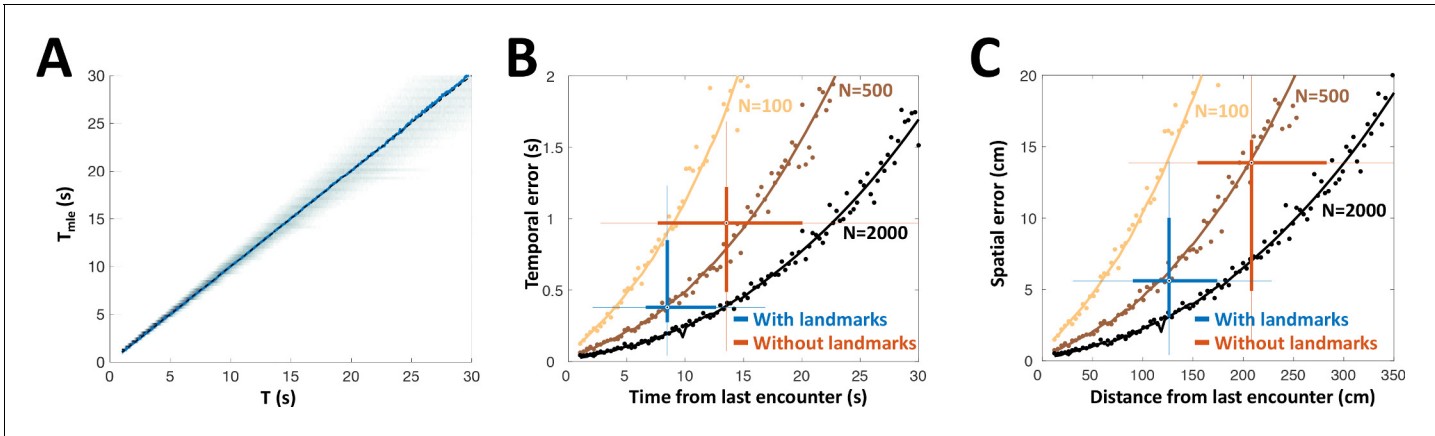

**Figure 8.** PG simulation explains animals' spatial and temporal behavioral accuracy. (**A**) Distribution of estimated time interval vs. actual time interval using a heterogeneous population (*N* = 500 cells). Estimation is approximately unbiased: the mean (blue line) follows the identity diagonal. (**B**) Temporal representation. Circles: mean error obtained numerically with 100 (yellow), 500 (brown) and 2000 (black) cells. Curves: CRLB computed analytically. Two-dimensional boxplots: distribution of behavioral temporal estimation errors vs. time from last encounter (blue: with landmarks, red: without landmarks, see ***Figure 7***). (**C**) Spatial representation. MLE temporal results were converted to distance by multiplying by the median velocity reported for navigating fish (12 cm/s) (***Jun et al., 2016***).
DOI: https://doi.org/10.7554/eLife.36769.020

that the validity of this computational analysis relies on our (admittedly simple) model correctly capturing the neuronal response dynamics. Finer details, such as the inter-dependency between the model's various parameters, were not taken into account in this study. However, it seems reasonable to suggest that the plentitude of PGl cells provides a large margin of error to accommodate such extensions.

## Discussion

Our results suggest the following space-to-time transformation scheme: PG derives a sequence of discrete novelty events (encounters) from OT activity. The remarkable history-independent adaptation process provides an accessible, accurate and unbiased representation of the time intervals between encounters. We have found visually-responsive PG cells displaying similar adaptation features (*Figure 5—figure supplement 2*), suggesting that this mechanism is implemented across multiple sensory modalities. The elimination of egocentric topographic information in PG – both in the response itself and in its adaptation – ensures that the encoding of time is invariant to the specific body part encountering the object. This temporal information is then transmitted to DL, which can use it to integrate the fish's swim velocity to obtain distance-traveled, a key allocentric variable. For fish, the necessary velocity information may be provided by the lateral-line system (*Chagnaud et al., 2007*; *Oteiza et al., 2017*); several lateral-line responsive PG units were encountered in our recordings (*Figure 3—figure supplement 1*). Finally, DL can combine the distance information with instantaneous heading-direction (vestibular system, *Straka and Baker, 2013*; *Yoder and Taube, 2014*) to yield the animal's allocentric spatial position (*Etienne and Jeffery, 2004*). A recent study in zebrafish suggests that DL neurons can indeed process temporal information on the long time-scales discussed here (*Cheng et al., 2014*). Our computational analysis demonstrates that the PG temporal information is sufficient to account for the spatial acuity displayed in behavioral studies of gymnotiform fish utilizing electrosensory information alone (*Figure 8*).

This space-to-time mechanism may shed light on the primitive basis of egocentric-to-allocentric transformations. Short-range sensing, used by ancestor species living in low-visibility environments, necessitated the perception of space through temporal sequences of object encounters. With the evolution of long-range sensory systems such as diurnal vision (*MacIver et al., 2017*), simultaneous apprehension of the spatial relations of environmental features became possible. The neural mechanisms implementing sequential (space-to-time) spatial inference and simultaneous spatial inference presumably both exist in mammals, for example, we can acquire a map of relative object locations by looking at a scene from afar, or by walking and sequentially encountering landmarks with our eyes closed. Whether the sequential or the simultaneous spatial-inference is more dominant may depend on the species (e.g., nocturnal or diurnal) and on context (e.g., open field or underground burrows). However, it is not clear whether sequentially versus simultaneously-acquired spatial knowledge is processed in a common circuit. Indeed, clinical case studies on the regaining of eyesight in blind humans indicate that sequential and simultaneous spatial perceptions are fundamentally different, and may therefore involve two distinct computations and neuronal pathways (*Sacks, 1995*). The population of thalamic neurons that we discovered may provide an essential component underlying one of these two major computations – the encoding of sequential temporal information – and we hypothesize that such neurons underlie sequential spatial learning in all vertebrates.

There is substantial evidence indicating that the pathway studied here indeed has parallels in other vertebrates, and specifically in mammals. PG's homology to posterior thalamic nuclei is supported by previously published anatomical and developmental findings (*Giassi et al., 2012a*; *Ishikawa et al., 2007*; *Mueller, 2012*), as well as by physiological (*Figure 1B,C*) and molecular (*Figure 1—figure supplement 1*) results presented here. The thalamic pulvinar nucleus is particularly similar to PG in that it receives direct tectal input (*Berman and Wurtz, 2011*). Its involvement in visual attention and saliency in primates (*Robinson and Petersen, 1992*) corresponds to PG's involvement in novelty detection (*Figure 2*). Moreover, pulvinar lesions are associated with saccadic abnormalities and deficits in the perception of complex visual scenes (*Arend et al., 2008*), suggesting a link to the sequential mode of spatial learning. *Komura et al. (2001)* demonstrated that posterior thalamic regions (including the rodent equivalent of the pulvinar) can implement interval timing computations over long time-scales (>>1 s); however, the mechanistic basis for these computations has not been identified (*Simen et al., 2011*) and potential contributions to path integration have not

been explicated. A recent paper (*Paton and Buonomano, 2018*) reviewed models of temporal encoding based on recurrent neural networks. Further studies will be required to determine whether the novel adaptation encoding mechanism in PG engages the downstream recurrent networks of DD and DL to produce refined estimates of the time interval between salient sensory and/or motor events. Finally, a recent study in rodents demonstrated spatially non-specific adaptation in VPM (*Jubran et al., 2016*), a posterior thalamic nucleus responding to vibrissal object encounters (*Yu et al., 2006*). Taken together, we hypothesize that thalamic space-to-time mechanisms akin to those presented here play an important role in mammalian sequential spatial learning, especially in nocturnal animals relying on sparse sensory cues (*Save et al., 1998*).

The telencephalic target of PG, DL, resembles the mammalian hippocampus not only in function, as revealed in lesion studies (*Rodríguez et al., 2002*), but also in development, gross circuitry and gene expression (*Elliott et al., 2017*). The role of the hippocampus in spatial learning and navigation is well established, and hippocampal neural correlates of allocentric spatial variables have been exquisitely described (*Barry and Burgess, 2014*; *Buzsáki and Moser, 2013*). There is also evidence for the importance of time coding in the mammalian hippocampus: 'Time cells' responsive to elapsed time have been reported and, in some cases, these cells also respond at specific spatial loci (*Eichenbaum, 2014*). Furthermore, a recent study on the representation of goals in the hippocampus found cells encoding the length/duration of the traveled path (*Sarel et al., 2017*). The mechanism we have found may therefore contribute to creating temporal coding in the hippocampus, not just in the context of egocentric-to-allocentric transformations but rather whenever expectations associated with specific time intervals need to be generated. It should be noted, however, that unlike DL's direct thalamic input via the PG bottleneck, the hippocampus receives sensory and motor information primarily via the cortex. Furthermore, multiple bi-directional pathways connect the mammalian sensory and motor cortical regions with the hippocampal network. Pinpointing the exact loci where egocentric-to-allocentric transformations may take place in the mammalian brain is therefore extremely challenging. We propose that this transformation is initiated in the mammalian thalamus where history-independent adaptation also encodes time between encounter events. Finally, we propose that this thalamic output contributes to the generation of an allocentric spatial representation in the mammalian hippocampus.

In this contribution, we propose a hypothesis about how gymnotiform fish, and perhaps vertebrates in general, generate their representation of position relative to the environment. Future experiments could test the predictions entailed by this hypothesis. Behaviorally, our model implies that the fish's sense of position is critically dependent on its last encounter with an object. Further behavioral studies of spatial learning could elucidate this relationship, for example, by manipulating the objects' arrangement relative to the navigation target. Combining these studies with chronic recordings of PG and its pallial targets in freely navigating fish will permit testing of our proposed space-to-time neural transformation scheme.

## Materials and methods

### Experimental model

All procedures were approved by the University of Ottawa Animal Care and follow guidelines established by the Society for Neuroscience. *Apteronotus leptorhynchus* fish (imported from natural habitats in South America) were kept at 28°C in community tanks. Fish were deeply anesthetized with 0.2% 3-aminobenzoic ethyl ester (MS-222; Sigma-Aldrich, St. Louis, MO; RRID: SCR_008988) in water just before surgery or tissue preparation.

### Surgical procedure for in-vivo recordings

Surgery was performed to expose the rostral cerebellum, lateral tectum and caudal pallium. Immediately following surgery, fish were immobilized with an injection of the paralytic pancuronium bromide (0.2% weight/volume), which has no effect on the neurogenic discharge of the electric organ that produces the fish's electric field. The animal was then transferred into a large tank of water (27°C; electrical conductivity between 100–150 µS cm$^{-1}$) and a custom holder was used to stabilize the head during recordings. All fish were monitored for signs of stress and allowed to acclimatize before commencing stimulation protocols.

## Neurophysiology

Custom made stereotrodes or tritrodes were made of 25 µm diameter Ni-Cr wire (California Fine Wires). Each electrode was manually glued to a pulled filamented glass pipette (P-1000 Micropipette Puller, Sutter Instrument, Novato, CA); the glass pipette provided mechanical rigidity that allowed advancing the tetrode to the deep-lying PG. Prior to recording, tetrode tips were gold-plated (NanoZ 1.4, Multi Channel Systems, Reutlingen, Germany) to obtain 200–300 kOhm impedance at 1 kHz. The electrode was positioned above the brain according to stereotaxic brain atlas coordinates (150–300 µm caudal to T26 and 800–1000 µm lateral to midline [*Maler et al., 1991*]), and lowered using a micropositioner (Model 2662, David Kopf Instruments, Tujunga, CA) while delivering visual and electrosensory stimuli. Tectal responses to such stimuli (*Bastian, 1982*) were usually detected twice, around 1200 µm and around 1900 µm ventral to the top of cerebellum (as expected from the curved shape of OT). The electrode usually then transversed nucleus Electrosensorius, producing weak multi-unit responses to electrocommunication stimuli around 2300–2500 µm (*Heiligenberg et al., 1991*). PG units were usually encountered between 2800 µm and 3400 µm ventral to the top of the cerebellum, and were easily identified due to their characteristic rapid spike bursts (*Figure 1B,C*). Differential extracellular voltage was obtained by using one stereotrode/tritrode channel as reference. This enabled near-complete cancellation of the electric organ discharge (EOD) interference.

We report on the responses of 84 PG neurons responsive to object motion; several motion protocols were used and the sample size for each protocol is mentioned in context. We also found PG cells responding to electrocommunication signals, mostly within the medial subdivision of PG (PGm). As expected from the sparse retinal input to OT (*Sas and Maler, 1986*), we recorded only a small number ($n = 19$) of PG cells responsive to visual input (*Figure 5—figure supplement 2*), mostly in more ventral portions of PG (*Figure 1—figure supplement 4*). In addition, we identified a small number of cells responsive to passive electrosensory (*Grewe et al., 2017*) (ampullary receptors, $n = 27$), acoustic ($n = 7$) and lateral line ($n = 7$, *Figure 3—figure supplement 1*) stimulation – but did not attempt to further characterize their coding properties.

Cell responses were initially manually tested with brass and plastic spheres. Cells responding to both were also tested with an electrically neutral gel ball made of 15% agarose in tank water to exclude lateral-line responses (*Heiligenberg, 1973*) (*Figure 3—figure supplement 1*). A plastic or brass sphere (1.21 cm diameter) was connected to an electromechanical positioner (Vix 250IM drive and PROmech LP28 linear positioner, Parker Hannifin, Cleveland, OH), which was pre-programmed for the appropriate motion sequence and initiated by outputs from our data acquisition software (Spike2, Cambridge Electronic Designs, Cambridge, UK). Typically, a trapezoidal velocity profile was used with 150 cm/s$^2$ acceleration and 5 cm/s peak velocity, and total distance in either direction of 8 cm (longitudinal) or 5 cm (transverse); due to technical limitations, each cell was recorded using only one direction of motion (longitudinal or transverse). For protocols involving motion in two axes (receptive field sampling, Oddball), the object was attached to a second electromechanical positioner (L12-100-50-12-P linear actuator, Firgelli Technologies, Ferndale, WA), which was mounted perpendicularly to the first one. In order to measure the receptive field (RF) size, we used one motor to repeatedly perform transverse object motion (towards and away from the fish) while a second motor was used to randomly change the longitudinal position between repetitions. This protocol was performed on a total of 27 cells. To check the spatial specificity of the adaptation process (i.e. if it is affecting only the body location experiencing the encounters or a whole-body effect), we performed a spatial oddball experiment: First, a series of N 'standard' encounters (in and out transverse object motion, N = 5–9) were given in rapid succession at one location (e.g. the head); the (N + 1)$^{th}$ encounter was given at a different location (e.g. the trunk) while maintaining the same time-interval between encounters (3–5 s, the second motor was used to quickly switch positions). Each such series was followed by a long recovery period ($\geq$ 30 s) and then repeated in the opposite direction. This was performed on a total of 14 cells, out of which six were found to be adapting (in either direction).

## Histology

In the initial experiments the location of PG was verified by preparing histological sections and locating the electrode track marks (*Figure 1—figure supplement 4*). After recordings were complete, the fish was deeply anesthetized using tricaine methanesulfonate (MS-222 0.2 g/L; Sigma-Aldrich,

St-Louis, MO) and transcardially perfused with 4% paraformaldehyde, 0.1% glutaraldehyde, and 0.2% picric acid in 0.1 M PBS pH 7.4. Brains were removed and incubated overnight in a solution of 4% paraformaldehyde, 0.2% picric acid, and 15% sucrose in 0.1 M PBS pH 7.4 at 4°C. Cryostat sections were cut at 25 μm in the transverse plane and mounted on Superfrost Plus glass slides (Fisher Scientific, Pittsburgh, PA). All sections were counterstained with green fluorescent Nissl reagent 1:300 (Molecular Probes, Eugene, OR; NeuroTrace 500/525 green-fluorescent Nissl Stain #N21480, RRID: SCR_013318) in PBS for 20 min at room temperature.

## Expression of T-type Ca$^{2+}$ channel α-subunits

Three adult male fish were anesthetized. Ice-cold ACSF was dripped on the head while the skull was removed and brains quickly removed and submerged in ice-cold ACSF. PG and DL brain regions were superficially located (*Maler et al., 1991*), identified, dissected out and stored at −20°C. Tissues were weighted and total RNA was purified using Trizol (Sigma-Aldrich) according to the manufacturer's recommendations. Contaminating genomic DNA was digested with DNAse1, total RNA was then precipitated overnight at −20°C, resuspended in nuclease-free water and quantified by spectroscopy. 300 ng of total RNA were used for first strand cDNA synthesis using the Maxima H Minus First Strand cDNA Synthesis Kit (K1681; ThermoFisher). PCR were made with the Taq polymerase (EP0402; ThermoFisher) according to manufacturer' recommendations for 35 cycles using the following primer pairs: G amplicon (301 bp), direct: 5'-CGACACCTTCCGCAAAATCG-3', reverse: 5'-AGCACAGACAGACCTCCGc-3'; H amplicon (338 bp), direct: 5'-GGGACGATTTCAGGGACAGG-3', reverse: 5'-CACTCGCAGCAGACGGAA-3'; I amplicon (355 bp), direct: 5'-TGGGATGAGATTGGAG TGAAAC-3', reverse: 5'-AGCGGACCAGCTTAATGACC-3'. Amplicons were then migrated on a 1% agarose Et-Br gel and photographed with a BIO-RAD Gel Doc System.

## Quantification and statistical analysis

### Preprocessing

Data were acquired using Spike2 (Cambridge Electronic Designs, Cambridge, UK) and analyzed in Matlab (MathWorks, Natick, MA). Single units were sorted offline by performing principal component and clustering analyses. Units were considered to be well-separated and to represent individual neurons, only if (a) their spike shapes were homogenous over time and did not overlap with other units or noise; and (b) the unit exhibited refractory period of > 1 ms in autocorrelation histograms (*Figure 1—figure supplement 3*).

### Burst analysis

Any spike preceded or followed by an ISI ≤ 8 ms was regarded as one participating in a burst. This was justified by observing the ISI first-return maps (*Ramcharan et al., 2005*), which showed characteristic clusters separated around 8 ms. To compute the burst fraction, the number of spikes participating in bursts was divided by the total number of spikes emitted by the unit.

### Information content

To determine the information the spiking response $r$ contains on object position $p$, the normalized mutual information was estimated: $U(r|p) = \frac{I(r,p)}{H(r)} = 1 - \frac{H(r|p)}{H(r)}$, where $I$ is the mutual information and $H$ is the information entropy. The conditional entropy $H(r|p)$ was calculated by binning the responses obtained in the RF sampling protocol (*Figure 2*) into 24 bins and calculating the conditional response (spike count) distribution in each bin. Note that $0 \leq U(r|p) \leq 1$, with $U(r|p) = 0$ when the spiking is completely independent of position, and $U(r|p) = 1$ when the firing is completely predictable given the object position.

### Response categorization

Motion response was categorized according to the epochs during which the cells' firing rate was at least twice the spontaneous rate (i.e. between motions). These epochs were defined as follows: motion onset: the first cm of motion in either direction; encounter: up to 2 cm from the skin; proximity: closer than 2 cm from the skin; motion offset: up to 200 ms after motion secession in either direction.

## Statistics

In all paired comparisons, the bootstrap method (5000 random redistributions without replacement) was employed to estimate statistical significance. Random permutations were used to evaluate significance of correlations (5000 random permutations without replacement). Sample sizes were determined by statistical requirements, aiming at confidence levels > 95%. No statistical methods were used to pre-determine sample sizes but our sample sizes are similar to those generally employed in the field. No randomization or blinding was used is this study.

## Dynamic adaptation model

The random interval protocol (*Figure 5A*) produced for each cell an interval (input) vector $\{T_n\}$ and a spike count response (output) vector $\{R_n\}$ (count computed for each unit in a time-window determined by its response type, see *Figure 3*). For some of the cells, a slow decline in response due to experimental instability was corrected by dividing the response time course by a least-squared fitted slow ( > 10 min) exponential decay. The model (*Figure 6*) assumes each neuron has a latent state variable $x$ with dynamics following each encounter given by

$$x_n = 1 - e^{-\frac{T_n}{\tau}}(1 - \beta x_{n-1}) \tag{1}$$

where $T_n$ is the $n^{th}$ time interval in the sequence, $\beta$ is the memory coefficient ($0 \leq \beta \leq 1$) and $\tau$ the recovery time-constant. Note that $x$ is always in the range [0,1]. Equation (1) is in fact the between-event, discrete-time solution of the standard resource-based depression model of Tsodyks and Markram (*Tsodyks and Markram, 1997*), where $\beta = 1 - U_{EI}$; we chose this formalization so that the extent of history-dependence in the adaptation dynamics will be emphasized (i.e., $\beta = 0$ signifies complete history independence and history dependence increases as $\beta \to 1$).

The neuron's firing parameter $\lambda_n$ at the $n^{th}$ encounter is $\lambda_n = [ax_n + c]_+$, where $a > 0$, $c \in R$ and $[\cdot]_+$ denotes non-negative rectification. Fitting of $a$ and $c$ was performed using linear least squares, while $\beta$ and $\tau$ were found by exhaustive search on a 2-D grid. Fitting was deemed successful if the parameter values generated by the model were correlated with the actual responses with $P < 0.05$ (random permutations, *Figure 6—figure supplement 1*). This was true for 15/33 cells. Note that when $\beta = 0$ ('history-independent' adaptation), we get:

$$\lambda_n = \left[a\left(1 - e^{-\frac{T_n}{\tau}}\right) + c\right]_+, \tag{2}$$

which corresponds to the solid blue line in *Figure 5B*.

Finally, $\{\tau_n\}$ were used as the parameters of Poisson random variables to generate stochastic spike-counts $\{R_n\}$, so that the number of spikes emitted at each encounter was distributed according to:

$$p(R_n = k) = \frac{\lambda_n^k e^{-\lambda_n}}{k!}, \quad k = 0, 1, 2, 3 \ldots \tag{3}$$

## Maximum-Likelihood estimation

Now we assume a population of N history-independent ($\beta = 0$) neurons, with $a_j, c_j$, and $\tau_j$ are the individual parameters of the $j^{th}$ neuron and $R_n^j$ is its response to the $n^{th}$ encounter. We also assume that $\{R_n^j\}_{j=1}^N$ are independent Poisson random variables (r.v.'s), with each r.v. distributed according to equation (3). The likelihood of this population response is therefore

$$L\left(T_n | \{R_n^j\}\right) = P\left(\{R_n^j\} | T_n\right) = \prod_{j=1}^{N} \frac{\left(\left[a_j\left(1 - e^{-\frac{T_n}{\tau_j}}\right) + c_j\right]_+\right)^{R_n^j} e^{-\left(\left[a_j\left(1 - e^{-\frac{T_n}{\tau_j}}\right) + c_j\right]_+\right)}}{R_n^j!},$$

and the log-likelihood is

$$l\left(T_n | \{R_n^j\}\right) = \sum_{j=1}^{N} R_n^j \log\left(\left[a_j\left(1 - e^{-\frac{T_n}{\tau_j}}\right) + c_j\right]_+\right) - \left[a_j\left(1 - e^{-\frac{T_n}{\tau_j}}\right) + c_j\right]_+ - \log\left(R_n^j!\right).$$

The Maximum-Likelihood Estimator of the last time interval is therefore obtained by finding the time interval $T_n$ that maximizes this likelihood:

$$T_n^{MLE} = \underset{T_n>0}{\mathrm{argmax}}\left(\sum_{j=1}^{N} R_n^j \log\left(\left[a_j\left(1-e^{-\frac{T_n}{\tau_j}}\right)+c_j\right]_+\right) - \left[a_j\left(1-e^{-\frac{T_n}{\tau_j}}\right)+c_j\right]_+\right).$$

This maximum was found numerically for each generated time interval.

## Homogeneous population model

To explore the coding properties of the model we used a population of N identical cells ($\tau_j = \tau$; $a_j = a$; $c_j = c$). Assuming for a moment $a\left(1-e^{-\frac{T_n}{\tau}}\right)+c>0$, the MLE for the population is the solution of:

$$\frac{dl(T_n)}{dT_n} = \sum_{j=1}^{N} a\left(-\frac{1}{\tau}e^{-\frac{T_n^{MLE}}{\tau}}\right)\left(1-\frac{R_n^j}{a\left(1-e^{-\frac{T_n^{MLE}}{\tau}}\right)+c}\right) = 0 \qquad (4)$$

$$N - \sum_{j=1}^{N}\left(\frac{R_n^j}{a\left(1-e^{-\frac{T_n^{MLE}}{\tau}}\right)+c}\right) = 0$$

$$a + c - \frac{1}{N}\sum_{j=1}^{N} R_n^j = a e^{-\frac{T_n^{MLE}}{\tau}}$$

$$T_n^{MLE} = -\tau \log\left(\frac{a+c-\frac{1}{N}\sum_{j=1}^{N} R_n^j}{a}\right) = \tau \log\left(\frac{a}{a+c-\frac{1}{N}\sum_{j=1}^{N} R_n^j}\right)$$

One can easily show that this solution indeed maintains $a\left(1-e^{-\frac{T_n^{MLE}}{\tau}}\right)+c>0$. Note that when $\frac{1}{N}\sum_{j=1}^{N} R_n^j > (a+c)$ then $T_n^{MLE}$ has no real solution (i.e., likelihood function has no finite maximum). In these cases, the estimator output is ignored.

Since for $f(X) = \log\left(\frac{1}{1-x}\right)$, $E(f(X)) \neq f(E(X))$ the MLE is biased. However, if we assume $T \ll \tau$, then we can approximate:

$$T_n^{MLE} \cong \tau\left(\frac{\frac{1}{N}\sum_{j=1}^{N} R_n^j - c}{a}\right)$$

for which:

$$E_{\{R_n^j\}}\left(T_n^{MLE}\right) \cong \tau\left(\frac{\lambda_n - c}{a}\right) = \tau\left(\frac{(ax_n+c)-c}{a}\right) = \tau\left(1-e^{-\frac{T_n}{\tau}}\right) \cong T_n$$

## Cramér-Rao lower bound

To compute the Fisher information for any time $T>0$, recall the identity:

$$I(T) = -E_{\{R^j\}}\left\{\frac{d^2 l(\{R^j\},T)}{dT^2}\right\} = -E_{\{R^j\}}\left\{\frac{d}{dT}\left(\sum_j -\frac{a_j}{\tau_j}e^{-\frac{T}{\tau_j}}\left(1 - \frac{R^j}{a_j\left(1-e^{-\frac{T}{\tau_j}}\right)+c_j}\right)\right)\right\},$$ where the sum is

restricted to those neurons with positive activity on that value of $T$, that is, $\{j|j \in [1:N] \wedge a_j\left(1 - e^{-\frac{T}{\tau_j}}\right) + c_j > 0\}$. Therefore:

$$I(T) = -E_{\{R^j\}}\left\{\sum_j \frac{a_j}{\tau_j^2}e^{-\frac{T}{\tau_j}}\left(1 - \frac{R^j}{a_j\left(1-e^{-\frac{T}{\tau_j}}\right)+c_j} - \frac{a_j R^j}{\left(a_j\left(1-e^{-\frac{T}{\tau_j}}\right)+c_j\right)^2}e^{-\frac{T}{\tau_j}}\right)\right\}$$

$$= -\sum_j \frac{a_j}{\tau_j^2}e^{-\frac{T}{\tau_j}}\left(1 - \frac{E\{R^j\}}{a_j\left(1-e^{-\frac{T}{\tau_j}}\right)+c_j} - \frac{a_j E\{R^j\}}{\left(a_j\left(1-e^{-\frac{T}{\tau_j}}\right)+c_j\right)^2}e^{-\frac{T}{\tau_j}}\right)$$

Since $R^j$ are Poisson r.v. with rate $\lambda_j = a_j\left(1 - e^{-\frac{T}{\tau_j}}\right) + c_j$:

$$I(T) = \sum_j \frac{a_j^2}{\tau_j^2\left(a_j\left(1-e^{-\frac{T}{\tau_j}}\right)+c_j\right)}e^{-\frac{2T}{\tau_j}}$$

For a homogeneous population (*Figure 6C–G*), this becomes:

$$I(T) = \frac{a^2 N}{\tau^2\left[a\left(1-e^{-\frac{T}{\tau}}\right)+c\right]_+}e^{-\frac{2T}{\tau}}$$

Therefore, the CRLB (assuming zero bias) becomes:

$$\text{Var}\{T^{\text{mle}}\} \geq \left(\sum_j \frac{a_j^2}{\tau_j^2\left(a_j\left(1-e^{-\frac{T}{\tau_j}}\right)+c_j\right)}e^{-\frac{2T}{\tau_j}}\right)^{-1}$$

## Simulation of bootstrap population

The set of parameter values obtained for all 15 fitted adapting cells (*a*, *c* and *τ*) were randomly drawn from to generate a bootstrap population of N cells (N=100, 500 and 2000). Gaussian noise with standard deviation of 25% was added to the parameters to obtain smoother distributions. The memory variable β was set to 0 for all cells to simplify simulations. We also assumed that spiking across the population was statistically independent with Poisson statistics given the last interval, that is $p\left(R_n^i, R_n^j | T_n\right) = p\left(R_n^i, | T_n\right)p\left(R_n^j, | T_n\right)$ where $R_n^i$ is the response of neuron *i* to time interval *n*. Random intervals were drawn in the range 1–30 s, and for each interval a vector of responses (spike counts) across the population $\left\{R_n^j\right\}_{j=1}^N$ was generated, where *N* is the population size.

## Analysis of behavioral data

Published spatial learning data (*Jun et al., 2016*) were used; methods used for the generation of these data are explained in detail in that study. Briefly, South American weakly electric fish (*Gymnotus* sp.) were trained to find food (a mealworm restrained to a suction cup) in complete darkness, at a specific location within a 150 cm diameter custom made circular arena. In experiments with landmarks, four acrylic objects (two square prisms, 5.6 and 9.0 cm/side and two cylinders, 7.6 cm and 10.2 cm diameter) were placed in fixed locations within the arena. In each daily session, each fish was given four trials to find the food. After a 12-session training stage, animal performance stabilized, and four test sessions were performed in which food was omitted in one randomly assigned 'probe' trial. Only data from these four last sessions were analyzed here. A total of four fish were used with landmarks and eight fish were used without landmarks. Fish behavior was video recorded and tracked as previously described (*Jun et al., 2014*).

The cruising time/distance from the last encounter (with either a landmark or the tank wall) to the location of the food is the epoch/trajectory the fish had to memorize in order to perform path

integration. This was measured in each of the 'food' trials (**Figure 7A,B**); the segment from the last encounter (3 cm from a landmark or 6 cm from the arena wall) to the detection of food (3 cm from food location) was found and the trajectory's total length and duration were computed. The spatial 'decoding' errors were obtained by measuring where the fish searched for the missing food in the probe trials (**Figure 7D,E**); the normalized histogram of the visiting frequency across space (heat map) was fitted with a two-dimensional Gaussian function:

$$P(x,y) = A \cdot exp\left(-\frac{1}{2(1-\theta^2)}\left(\frac{(x-\mu_x)^2}{\sigma_x^2} + \frac{(y-\mu_y)^2}{\sigma_y^2} - \frac{2\theta(x-\mu_x)(y-\mu_y)}{\sigma_x\sigma_y}\right)\right)$$

where $\mu_x$ and $\mu_y$ are mean parameters; $\sigma_x$ and $\sigma_y$ are variance parameters; $A$ is a gain parameter and $\theta$ is the cross correlation parameter. The distance between the Gaussian center ($\mu_x$, $\mu_y$) (white 'x' mark) and the food location (white '+' mark) is the spatial error; dividing this error by the median velocity in the trial produces the temporal error.

## Acknowledgments

We thank Bill Ellis for technical support and Nachum Ulanovsky and Nate Sawtell for comments and suggestions. This research was supported by NSERC Grant 121891–2009 (to AL), NSERC Grant 147489–2017 (to LM) and Canadian Institutes of Health Research Grant 49510 (to LM and AL).

## Additional information

### Funding

| Funder | Grant reference number | Author |
| --- | --- | --- |
| Natural Sciences and Engineering Research Council of Canada | 121891-2009 | André Longtin |
| Canadian Institutes of Health Research | 49510 | André Longtin Len Maler |
| Natural Sciences and Engineering Research Council of Canada | 147489-2017 | Len Maler |

The funders had no role in study design, data collection and interpretation, or the decision to submit the work for publication.

### Author contributions

Avner Wallach, Data curation, Software, Formal analysis, Investigation, Visualization, Methodology, Writing—original draft, Writing—review and editing, Performed the electrophysiological experiments, data analysis and modeling; Erik Harvey-Girard, Investigation, Writing—review and editing, Performed PCR experiments; James Jaeyoon Jun, Data curation, Writing—review and editing, Performed behavioral experiments; André Longtin, Supervision, Funding acquisition, Project administration, Writing—review and editing; Len Maler, Conceptualization, Resources, Supervision, Funding acquisition, Project administration, Writing—review and editing

### Author ORCIDs

Avner Wallach (iD) http://orcid.org/0000-0003-2345-2942
André Longtin (iD) http://orcid.org/0000-0003-0678-9893
Len Maler (iD) http://orcid.org/0000-0001-7666-2754

### Ethics

Animal experimentation: All procedures were approved by the University of Ottawa Animal Care and follow guidelines established by the Society for Neuroscience (approved protocol number: CMM-2897)

Decision letter and Author response
Decision letter https://doi.org/10.7554/eLife.36769.025
Author response https://doi.org/10.7554/eLife.36769.026

## Additional files

### Supplementary files
• Transparent reporting form
DOI: https://doi.org/10.7554/eLife.36769.021

### Data availability

Datasets and analysis files have been deposited in Columbia University's Academic Commons repository (https://doi.org/10.7916/D86Q3F7S).

The following dataset was generated:

| Author(s) | Year | Dataset title | Dataset URL | Database and Identifier |
|---|---|---|---|---|
| Wallach A, Harvey-Girard E, Jun JJ, Longtin A, Maler L | 2018 | Data for: A time-stamp mechanism may provide temporal information necessary for egocentric to allocentric spatial transformations | https://doi.org/10.7916/D86Q3F7S | Columbia Academic Commons, 10.7916/D86Q3F7S |

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
