## [Decision Letter]

Thank you for submitting your article "A novel time-stamp mechanism transforms egocentric encounters into an allocentric spatial representation" for consideration by *eLife*. Your article has been reviewed by three peer reviewers, including Catherine Emily Carr as the Reviewing Editor and Reviewer #1, and the evaluation has been overseen by Eve Marder as the Senior Editor. The following individuals involved in review of your submission have agreed to reveal their identity: Matthew A Wilson (Reviewer #2).

The reviewers have discussed the reviews with one another and the Reviewing Editor has drafted this decision to help you prepare a revised submission.

Summary:

This paper contains the first recordings from neurons in the periglomerular complex (PG) in weakly electric fish. This structure receives input from the optic tectum and projects to the dorsolateral pallium (DL). DL is hypothesized to be the site of spatial memory, and thus its input from this thalamic complex is important. The authors have recorded from PG, and show that, despite the topographic nature of the input from the optic tectum, responses are non-topographic. They use a model to support the hypothesis that what occurs in PG is a temporal representation of spatial sequences.

Essential revisions:

The reviewers were divided about your manuscript. Two felt there was sufficient merit in this being the first report of activity in the periglomerular complex, while a third felt that both the analysis and the presentation of data were insufficient. All three reviewers agreed that the conclusions of the paper were not satisfactorily supported by the data. From the Senior Editor: "We note an increasing tendency in submitted manuscripts for authors to "overhype" their results to "sell" their work. We strongly encourage you to let your data speak for themselves and to present the data in a way that the reader can see exactly what you have done and why."

The manuscript lacks the information required to assess the strength and significance of many statements, with a need for substantial improvement in the presentation of results.

The presentation and analysis of the electrophysiological data require careful revision, potentially with a table, to show which cells were evaluated with which stimuli, and where they were located. In many cases the numbers mentioned appear inconsistent. More details are provided below.

The simulations are limited and only demonstrate plausibility of the proposed mechanisms.

The analysis of the behavioral data was not explained with sufficient clarity.

Introduction:

The opening paragraph of the Introduction states that "neural mechanisms underlying the transformation of the egocentric sensory and motor information streams into an allocentric representation.… are completely unknown". This is incorrect. Even though much is still unknown, a lot has been learned about the emergence of an allocentric representation of position and especially heading – in rodents and in other organisms. Some notable recent examples include the discoveries related to representation of orientation in *Drosophila*; or the recent work by Peyrache et al., reporting on the existence of a conjunctive representation of allocentric heading and egocentric proximity to borders, which may serve as a building block for the allocentric border cells observed in the entorhinal cortex.

There are many assertions throughout the manuscript that appear unsupported. For example, "making PG a feed-forward information bottleneck between egocentric and allocentric spatial representations" could be reconsidered. Just because temporal information, combined with a speed signal, can permit accurate path-integration, does not mean that it does. The authors should critically review all assertions to differentiate between what is shown and what is hypothesized.

Results section:

Further information is required about the PG responses and their analysis in the manuscript. Questions raised in the reviews are summarized below. Part of the confusion experienced when reading the manuscript may emerge from the separation of figures and supplementary figures. These should be integrated to support a logical flow of results. For example, why does Figure 4—figure supplement 2 contains navigation behavior and lateral line physiology?

– What is the relationship between a cell's properties as shown in Figure 2A-C to those shown in Figure 2D-F, to those shown in Figure 3? What are the parameters that characterize the population of time-interval encoding cells, as extracted from the procedures described in the Materials and methods section?

How were the cells, shown in Figure 2 and discussed in subsection “PG cells respond to object encounters”, classified into the different categories?

It is unclear how the various sample pools of cells were selected and how they overlap. 84 cells were recorded, but across how many animals? I assume each animal was implanted with single electrodes (stereo or tri -trodes), and that only single penetrations were made for each animal although this is not stated explicitly.

Of the 84 cells, it is reported that 27 had receptive fields mapped in Figure 1. They then describe 28 cells tested with longitudinal motion. I assume that these are a separate group of cells measured in a separate group of animals, although again not explicitly stated. This is followed by description of 40 cells showed looming-receding responses described in Figure 2. This is slightly confusing given that the receptive field mapping of the 27 cells in Figure 1 used a looming-receding protocol. Did the 40 cells shown in Figure 2 simply go through a more extensive mapping protocol allowing the different detection types (proximity, encounter, change) to be identified (although no such description is given in the methods)? Again, is this a separate pool of cells in a separate group of animals?

Given that the numbers here don't quite add up (27+40+28 = 95) there is something that I am missing. Perhaps there is some overlap between the 27 receptive field mapped cells and the 40 looming-receding cells, and then the question is why the subset?

Figure 3 then describes 33 cells subjected to repeated motion protocols. Again, unclear how this pool of cells relates to the other pools.

Perhaps a summary table in the supplemental information listing the all of the animal/cells/protocol would clarify things.

It would be useful to include data regarding the receptive fields as a function of body position for the units shown in Figure 1 to get a sense of the response distributions along the body.

If the cells are drawn from different animals/recordings, how confident are the authors that the 3 topographic cells shown are drawn from the same pool as the non-topographic and are not the result of sampling from a different site due to variation in electrode placement across implants. This is not essential to the overall interpretation, but it would be important to know whether this reflects an accurate estimate of the relative representational heterogeneity in PG.

I did not understand the terminology used in the manuscript. Most of the cells described in PG exhibit invariance to the heading of an object relative to the animal, but they are selective to the distance of an object from the animal (and they do not acquire selectivity to the animal's heading relative to the environment). Why, then, claim that egocentric spatial information is abolished in PG?

In addition, how do the results in the manuscript show that the time interval encoding observed in PG produces an allocentric spatial representation, as announced in the title? The results of the manuscript only hint at the possibility that the activity in PG might serve as an input to this computation.

The conclusions drawn from the model should, in my opinion, be taken with a great deal of caution, because of the assumptions that were made: first, the memory variable was set to zero (is this justified based on the fits?) Even more importantly, the model assumes independent noise in the different neurons. There are various reasons why this might be incorrect, possibly leading to a greatly reduced ability to decode the interval duration from a large population: one of them is correlated stochasticity. Another reason is that the activity might depend on some latent variables other than the history of time intervals. Overall, I am not convinced that the population activity in PG can be probed to generate a robust readout of the time interval between encounters.

I would also like to comment that the computational analysis in the paper is not strong. I appreciate that the model of neural response is simple but the fit of the data to this simple model is not demonstrated convincingly. The neural readout analysis suffers from technical issues (see below), but even more importantly it relies on assumptions that may be incorrect, and these limitations are not acknowledged or discussed.

The assessment of readout precision is performed by simulating spiking activity (based on the model) and application of a maximum likelihood (ML) decoder. This is unnecessary. Given the assumptions of independent Poisson noise and no history dependence, it is possible to evaluate the Fisher Information analytically, which would clarify how the results scale with various parameters. Considering the large number of cells postulated, this analysis is expected to provide precise agreement with the ML simulation results. However, I also expect to see some assessment for how the observed distribution of the history dependence parameter *β* might affect decoding precision. This aspect of the analysis may be more difficult to achieve with a full analytical approach, hence simulations can be helpful. A simple estimator that ignores the history dependence might be significantly influenced by the history dependence, as this is a source of variability that is shared across the population. Perhaps a more sophisticated decoder might do better, but this would require a reasonable proposal for implementation by neural circuitry. A decoder that takes into account the history dependence may need to maintain memory of the response from the previous encounter, and it's important to understand whether this is necessary.

The analysis of the behavioral data is not explained with sufficient clarity (for example, what are the parameters theta and A mentioned in subsection “Analysis of behavioral data”?). How the model of neural readout relates to the behavior is even less clear. Finally, in in subsection “PG activity explains path integration acuity” it is stated that "the number of PGI cells is sufficient to attain the observed behavioral precision, even when additional encoding errors e.g. in heading and velocity estimation are taken into account". Where is this demonstrated?

3) Title: The title is a bit misleading given that the egocentric-allocentric transformation is not explicitly demonstrated but rather suggested through simulation. Perhaps qualifying it with "A novel time-stamp mechanism could transform egocentric encounters into allocentric spatial representations".

[Editors' note: further revisions were requested prior to acceptance, as described below.]

Thank you for submitting your article "A time-stamp mechanism may provide temporal information necessary for egocentric to allocentric spatial transformations" for consideration by *eLife*. Your article has been reviewed by Eve Marder as the Senior Editor, a Reviewing Editor, and two reviewers. The following individual involved in review of your submission has agreed to reveal his identity: Matthew A Wilson (Reviewer #2).

The reviewers have discussed the reviews with one another and the Reviewing Editor has drafted this decision to help you prepare a revised submission.

Summary:

This paper contains the first recordings from neurons in weakly electric fish that project to the dorsolateral pallium (DL). DL is hypothesized to be the site of spatial memory, and this input is hypothesized to transform egocentric encounters into allocentric spatial representations.

Essential revisions:

1) There remains concern about how well the neural response model captures the measured neural responses. Figure 6—figure supplement 1 demonstrates that the predictions of the model and the measurements are positively correlated, and that this correlation is statistically significant; but this on its own is a relatively weak statement, and it is difficult to interpret the reported values of correlation coefficients. We would like to see a word of caution that the analysis relies on an assumption that the model correctly captures the neural responses.

2) The mathematical analysis of temporal encoding precision assumes that *β* = 0, and accurate readout using the naive ML estimator requires approx. *β* < 0.2. Hence, it's relevant to know, what are the characteristic properties of cells with zero or small *β*: one can imagine a scenario in which these particular cells have very small gains, or short time constants, and are thus less useful for the computation than expected from panels A-C.

3) The authors propose a hypothesis about how animals generate their representation of position relative to the environment, and could discuss how this prediction might be tested in future experiments, either behaviorally or in terms of neural activity in other brain areas. Behaviorally, the model implies that the animal's sense of position is critically dependent on the last encounter with an object. Does this make sense? Another interesting prediction is that there may be a cutoff in the durations of swimming without encounters, over which the animal can estimate its position – determined by the distribution of adaptation time constants.

[Editors' note: further revisions were requested prior to acceptance, as described below.]

Thank you for submitting your article "A time-stamp mechanism may provide temporal information necessary for egocentric to allocentric spatial transformations" for consideration by *eLife*.

We'd like to accept your paper, but first suggest you revise the paragraph excerpted below because it does not really address the reviewer's question.

What we asked for in point 3 was "The authors propose a hypothesis about how animals generate their representation of position relative to the environment, and could discuss how this prediction might be tested in future experiments, either behaviorally or in terms of neural activity in other brain areas. Behaviorally, the model implies that the animal's sense of position is critically dependent on the last encounter with an object. Does this make sense?"

You added a paragraph to the Discussion section, discussing several possible lines of behavioral and physiological inquiry for future studies. We would like a shorter reply that is more to the point in addressing this "Behaviorally, the model implies that the animal's sense of position is critically dependent on the last encounter with an object. Does this make sense?"

---

## [Author Response]

Summary:This paper contains the first recordings from neurons in the periglomerular complex (PG) in weakly electric fish. This structure receives input from the optic tectum and projects to the dorsolateral pallium (DL). DL is hypothesized to be the site of spatial memory, and thus its input from this thalamic complex is important. The authors have recorded from PG, and show that, despite the topographic nature of the input from the optic tectum, responses are non-topographic. They use a model to support the hypothesis that what occurs in PG is a temporal representation of spatial sequences.Essential revisions:The reviewers were divided about your manuscript. Two felt there was sufficient merit in this being the first report of activity in the periglomerular complex, while a third felt that both the analysis and the presentation of data were insufficient. All three reviewers agreed that the conclusions of the paper were not satisfactorily supported by the data. From the Senior Editor: "We note an increasing tendency in submitted manuscripts for authors to "overhype" their results to "sell" their work. We strongly encourage you to let your data speak for themselves and to present the data in a way that the reader can see exactly what you have done and why."

We absolutely agree, and the manuscript was revised to clearly differentiate between findings, conjectures and hypotheses. See below our detailed response.

The manuscript lacks the information required to assess the strength and significance of many statements, with a need for substantial improvement in the presentation of results.The presentation and analysis of the electrophysiological data require careful revision, potentially with a table, to show which cells were evaluated with which stimuli, and where they were located. In many cases the numbers mentioned appear inconsistent. More details are provided below.

We agree, and a new supplementary table was prepared (now Figure 1—figure supplement 2), showing which protocols were used on each of the neurons reported in this study, which were used on each animal reported in this study, and the sum total of cells per protocol. The distribution of depths in which the units were found was added to the figure supplement showing anatomy and histology (now Figure 1—figure supplement 4).

The simulations are limited and only demonstrate plausibility of the proposed mechanisms.

We agree and following the reviewers’ comments we added a new section exploring the model both analytically and numerically (subsection “PG cells convey readily accessible temporal information”, see our detailed response below). While our more extensive analyses strengthen our claim that our proposed mechanism is important for spatial memory, we cannot prove it. In the Discussion section, we now cite a recent review that suggests that recurrent neural networks can encode duration intervals. PG projects to the recurrent network of DL and we now note that our suggested mechanism may only be part of more extensive circuitry that converts encounter intervals to a spatial map.

The analysis of the behavioral data was not explained with sufficient clarity.

We agree, and the section describing the behavioral results was thoroughly revised.

Introduction:The opening paragraph of the Introduction states that "neural mechanisms underlying the transformation of the egocentric sensory and motor information streams into an allocentric representation.… are completely unknown". This is incorrect. Even though much is still unknown, a lot has been learned about the emergence of an allocentric representation of position and especially heading – in rodents and in other organisms. Some notable recent examples include the discoveries related to representation of orientation in Drosophila; or the recent work by Peyrache et al., reporting on the existence of a conjunctive representation of allocentric heading and egocentric proximity to borders, which may serve as a building block for the allocentric border cells observed in the entorhinal cortex.

We thank the Editor and reviewers for pointing out these recent papers. We now add a brief description of the Peyrache et al., paper (mouse) and a Selig et al., paper (*Drosophila*); both papers do make important advances in our understanding of the generation of allocentric representations. We also note that, in the case of the mouse, the circuitry involved is so complicated that a fully mechanistic model is still out of reach. We hope that our brief summary (Introduction) of very complex papers is sufficient to alert readers to work in this field.

There are many assertions throughout the manuscript that appear unsupported. For example, "making PG a feed-forward information bottleneck between egocentric and allocentric spatial representations" could be reconsidered. Just because temporal information, combined with a speed signal, can permit accurate path-integration, does not mean that it does. The authors should critically review all assertions to differentiate between what is shown and what is hypothesized.

We fully agree. We now provide much more detailed information supporting our contention that the information related to object motion is converted to a spatial map by the pallial circuitry (Introduction). We cannot prove that the temporal information conveyed by PG to pallium is the key input and so clearly state that this is a hypothesis based on our new data and previous papers.

Results section:Further information is required about the PG responses and their analysis in the manuscript. Questions raised in the reviews are summarized below. Part of the confusion experienced when reading the manuscript may emerge from the separation of figures and supplementary figures. These should be integrated to support a logical flow of results. For example, why does Figure 4—figure supplement 2 contains navigation behavior and lateral line physiology?

We agree and re-organized the figures to support a logical and easy-to-follow flow of results. The following table summarizes the changes made:

New figure #ContentTaken from old figureFigure 1Introductory figure: neural sensory pathways, burstingFigure 1A-CFigure 1—figure supplement 1T-type Ca^2+^ expressionFigure 1—figure supplement Figure 1Figure 1—figure supplement 2Dataset tablesNewFigure 1—figure supplement 3Spike sortingFigure 1—figure supplement Figure 2Figure 1—figure supplement 4Anatomy and histologyFigure 1—figure supplement Figure 3 + New depth histogramFigure 2Receptive fieldsFigure 1D-GFigure 1—figure supplement 1RFs across populationNewFigure 3Novelty detectionFigure 2Figure 3—figure supplement 1Lateral line resultsFigure 4—figure supplement Figure 2Figure 3—figure supplement 2Electric image modelFigure 2—figure supplement Figure 1Figure 4Response to Brass/PlasticFigure 2—figure supplement Figure 2Figure 5AdaptationFigure 3Figure 5—figure supplement 1Response to periodic motionFigure 3—figure supplement Figure -1Figure 5—figure supplement 2Adaptation to visual stimuliFigure 3—figure supplement Figure 3Figure 6Model and MLEFigure 3—figure supplement Figure 2 + newFigure 6—figure supplement 1Empirical parameter distributionsNewFigure 7Behavioral experimentsFigure 4—figure supplement Figure 1Figure 8MLE of heterogeneous populationFigure 4

What is the relationship between a cell's properties as shown in Figure 2A-C to those shown in Figure 2D-F, to those shown in Figure 3?

Due to technical limitations, each cell was recorded using only one direction of motion-longitudinal or transverse (this is now stated explicitly in Materials and methods section), and so the relationship between the response properties in these two axes is currently unknown. All response types were observed in adapting cells (e.g. of the 11 tested with radial motion, 8 displayed proximity, 6 encounter and 4 motion change responses, with 4 displaying more than one response type). However, the numbers are too small to say anything conclusive and so we believe that including this breakdown in the paper will not contribute to the readers’ understanding.

What are the parameters that characterize the population of time-interval encoding cells, as extracted from the procedures described in the Materials and methods section?

A new figure supplement was added (Figure 6—figure supplement 1) showing the empirically obtained CDFs of all model parameters.

How were the cells, shown in Figure 2 and discussed in subsection “PG cells respond to object encounters”, classified into the different categories?

In the initial submission, categorization was performed manually, by observing the PSTH of the neuronal response. Following the reviewer’s comment, we performed this analysis with precisely defined criteria (now described in Materials and methods section). While this slightly changed the cell counts, it did not alter any of the paper’s main claims.

It is unclear how the various sample pools of cells were selected and how they overlap. 84 cells were recorded, but across how many animals? I assume each animal was implanted with single electrodes (stereo or tri -trodes), and that only single penetrations were made for each animal although this is not stated explicitly.Of the 84 cells, it is reported that 27 had receptive fields mapped in Figure 1. They then describe 28 cells tested with longitudinal motion. I assume that these are a separate group of cells measured in a separate group of animals, although again not explicitly stated. This is followed by description of 40 cells showed looming-receding responses described in Figure 2. This is slightly confusing given that the receptive field mapping of the 27 cells in Figure 1 used a looming-receding protocol. Did the 40 cells shown in Figure 2 simply go through a more extensive mapping protocol allowing the different detection types (proximity, encounter, change) to be identified (although no such description is given in the methods)? Again, is this a separate pool of cells in a separate group of animals?Given that the numbers here don't quite add up (27+40+28 = 95) there is something that I am missing. Perhaps there is some overlap between the 27 receptive field mapped cells and the 40 looming-receding cells, and then the question is why the subset?Figure 3 then describes 33 cells subjected to repeated motion protocols. Again, unclear how this pool of cells relates to the other pools.Perhaps a summary table in the supplemental information listing the all of the animal/cells/protocol would clarify things.

We fully agree, and new supplementary tables were prepared as the reviewer suggested (now Figure 1—figure supplement 2). These tables show which protocols were used on each of the neurons reported in this study, which were used on each animal reported in this study, and the sum total of cells per protocol is displayed. We also list the number of penetrations, recording sites and single units in each experiment.

It would be useful to include data regarding the receptive fields as a function of body position for the units shown in Figure 1 to get a sense of the response distributions along the body.

We agree, and a figure depicting these data was added (now Figure 2—figure supplement 1).

If the cells are drawn from different animals/recordings, how confident are the authors that the 3 topographic cells shown are drawn from the same pool as the non-topographic and are not the result of sampling from a different site due to variation in electrode placement across implants. This is not essential to the overall interpretation, but it would be important to know whether this reflects an accurate estimate of the relative representational heterogeneity in PG.

The reviewer raises a valid question- do the few topographic cells found represent an anatomically distinct sub-structure or are they evenly distributed in PGl? Not all of the 3 cells found were encountered in the same place (2 were recorded in a single location while the third in another fish). We feel that it is unlikely that we missed any subnucleus in PG because we attempted to thoroughly explore the entire 3D spatial span of PG:

For the dorsal-ventral axis, we advanced from the edge of nE to the ventral edge of the brain. nE can be readily identified via the responses of its neurons to only electrocommunication signals (we cite the relevant papers). We found object motion related responses throughout most of lateral PG (PGl); towards its ventral edge we found a small number of neurons responsive to visual input only (reported and illustrated in a Supplementary figure); after this, we exited the brain.

We routinely sampled the full medio-lateral extent finding numerous cells responsive to electrocommunication signals medially (within PGm, not reported in this study); these cells were deep were located deeper than nE and responded differently than nE cells.

Finally, we sampled rostro-caudally; rostrally we found cells responsive to exogenous electric signals that are encoded by ‘ampullary receptors’ (PGr, not reported) followed by the cells responsive to object motion caudal to this rostral PG region; as we progressed caudally in our search we finally ended up with no response to any stimulus we presented and we assumed that we were out of PG.

Hence it is unlikely that one of the currently recognized large subdivisions of PG (PGl, PGm or PGr) is entirely dedicated to topographic representation. We cannot reject the possibility of a smaller such structure, but with the small number of topographic cells found, nothing can be said for certain one way or another. We added a sentence in the manuscript relating to this open question (subsection “Topographic spatial information is abolished in PG”).

I did not understand the terminology used in the manuscript. Most of the cells described in PG exhibit invariance to the heading of an object relative to the animal, but they are selective to the distance of an object from the animal (and they do not acquire selectivity to the animal's heading relative to the environment). Why, then, claim that egocentric spatial information is abolished in PG?

We agree with the reviewer on this point- indeed, the variance in sensitivity to object distance across the PG population provides egocentric information on object proximity. The coding of this information, however, is fundamentally different than that of ELL (the only region so far where this aspect was studied, see below). We now acknowledge this point in the relevant section in the text (subsection “PG cells respond to object encounters”). To clarify our terminology, we now use wherever possible the term *topographic egocentric information,* which relates to the directional egocentric mapping that characterizes all upstream electrosensory regions (ELL, nP, TS, OT) and is largely abolished in PG.

Explanation of the difference:

In ELL, coding for electrosensory object location with respect to the body surface (nose to tail, back to belly) is via a classic topographic projection from electroreceptors to ELL. Coding for distance from the body is via a standard firing rate code – firing rate of ON/OFF ELL pyramidal cells increases smoothly as a metal/plastic sphere is moved closer to the fish (the responses of tectal cells to such ‘looming’ signals has not been studied). This response of ELL cells is very different from that of PG cells. PG cells do not respond with (in)de-creasing firing rate as objects are moved closer (looming) or further (receding) from the body. Instead they respond in three different modes and typically ignore the metal/plastic stimulus dimension: they discharge when the objects are very close to the body (1 cm), when they very far away (4 cm) or when they suddenly begin to move at any location within their receptive range. In other words, this is not a simple egocentric map of distance from body; we do not understand and have no hypothesis on how these responses are used by the PG target in pallium (DL) and so did not expand on our simple description of these responses. We feel that this change in representation of ‘distance from body’ is off-topic for our manuscript and so did not elaborate on this point in the discussion.

In addition, how do the results in the manuscript show that the time interval encoding observed in PG produces an allocentric spatial representation, as announced in the title? The results of the manuscript only hint at the possibility that the activity in PG might serve as an input to this computation.

We agree and changed the wording to indicate that we hypothesize that the temporal representation described in our Results section, is used for the purpose of generating an allocentric spatial representation, e.g. the Title and Introduction.

The conclusions drawn from the model should, in my opinion, be taken with a great deal of caution, because of the assumptions that were made: first, the memory variable was set to zero (is this justified based on the fits?)

We agree with the reviewer that the effect of the memory parameter is indeed an important issue to explore- see below where we show results that suggest how non-zero memory leaves our basic conclusions unchanged.

Even more importantly, the model assumes independent noise in the different neurons. There are various reasons why this might be incorrect, possibly leading to a greatly reduced ability to decode the interval duration from a large population: one of them is correlated stochasticity. Another reason is that the activity might depend on some latent variables other than the history of time intervals. Overall, I am not convinced that the population activity in PG can be probed to generate a robust readout of the time interval between encounters.

We agree that this is an important assumption. Testing for the validity of this assumption is not feasible using our dataset. However, one relatively straightforward solution to this complication relies on the large sub-population of *non-adaptive* cells (55% of cells tested with the random interval protocol). Assuming that these cells are affected by the same correlations/latent variable, their response may provide an easy indicator to the underlying state of the system. A downstream network may thus use this information to correct its temporal estimation derived from the adaptive sub-population. We are currently computationally exploring the possible roles of these non-adaptive cells. While we can demonstrate this mechanism here with additional simulations, we feel that this extends beyond the scope of this contribution and prefer addressing this issue in a future theoretical paper. In the revised manuscript, we now restate the independence assumption explicitly and suggest the non-adaptive population as a possible solution (subsection “PG cells convey readily accessible temporal information”).

I would also like to comment that the computational analysis in the paper is not strong. I appreciate that the model of neural response is simple but the fit of the data to this simple model is not demonstrated convincingly.

Figure 6—figure supplement 1 now includes a panel showing the correlations of the data with the model, all of which were statistically significant (p<0.05, random permutations).

The neural readout analysis suffers from technical issues (see below), but even more importantly it relies on assumptions that may be incorrect, and these limitations are not acknowledged or discussed.The assessment of readout precision is performed by simulating spiking activity (based on the model) and application of a maximum likelihood (ML) decoder. This is unnecessary. Given the assumptions of independent Poisson noise and no history dependence, it is possible to evaluate the Fisher Information analytically, which would clarify how the results scale with various parameters. Considering the large number of cells postulated, this analysis is expected to provide precise agreement with the ML simulation results.

We agree with the reviewer that a more thorough exploration of the computational model and the readout mechanism is required. We added a new section, after the adaptation results and before the behavioral ones (subsection “PG cells convey readily accessible temporal information”), where we explore the Fisher information issue following the reviewer’s suggestions. We use a model of a homogeneous population (identical parameters for all cells) to demonstrate the scaling of estimation error with the various parameters. While the MLE is only approximately unbiased for short time-intervals, it indeed demonstrates perfect agreement with the Cramer-Rao Lower Bound across a wide range of stimuli. We then compute the CRLB also for the heterogeneous population used in comparison with the behavioral data.

However, I also expect to see some assessment for how the observed distribution of the history dependence parameter β might affect decoding precision. This aspect of the analysis may be more difficult to achieve with a full analytical approach, hence simulations can be helpful. A simple estimator that ignores the history dependence might be significantly influenced by the history dependence, as this is a source of variability that is shared across the population. Perhaps a more sophisticated decoder might do better, but this would require a reasonable proposal for implementation by neural circuitry. A decoder that takes into account the history dependence may need to maintain memory of the response from the previous encounter, and it's important to understand whether this is necessary.

We agree with this important comment. First, we note that 33% (5/15) of the adapting cells were best modeled with no memory at all, i.e. *β*=0 (this was not stated anywhere in the previous version of this paper; we now added Figure 6—figure supplement 1, which shows the empirically obtained CDFs of all parameters). Thus, even the strictly memoryless sub-population appears to be quite large. However, the reviewer is correct in that the effects of non-zero *β* are important to analyze. We therefore checked the scaling of bias and RMSE when *β* is increased for the entire homogeneous population. The MLE was constructed by fitting the model’s activity with a zero memory model (i.e. assuming the data were generated by a population with *β*=0). Despite the population-wide history-dependence introduced, the mean bias of the model remained negligible up to *β*=0.5, and the RMSE up to *β*=0.2. Thus, the time intervals may be decoded with a simple memoryless-based MLE using most of the adapting PG units. (median *β*=0.12) These results are now explained in the new modeling and decoding section (subsection “PG cells convey readily accessible temporal information”) and in Figure 6H.

The analysis of the behavioral data is not explained with sufficient clarity (for example, what are the parameters theta and A mentioned in subsection “Analysis of behavioral data”?).

We agree; explanation of the experimental procedures and rationale (subsection “PG activity corresponds to observed path integration acuity”), as well as the figure illustrating these procedure (Figure 7), are now in the main text. We also added the fitted Gaussian expression to the Materials and methods section.

How the model of neural readout relates to the behavior is even less clear.

We agree. To make this clearer and easier to follow, the neural readout mechanism and the behavioral results are now explained separately in new sections, each with its own figure (Figure 6 and Figure 7). Only then are the two combined in the final Results section and Figure 8. We also expand on our description of the behavioral results and how they related to our data and analyses (subsection “PG activity corresponds to observed path integration acuity”).

Finally, in in subsection “PG activity explains path integration acuity” it is stated that "the number of PGI cells is sufficient to attain the observed behavioral precision, even when additional encoding errors e.g. in heading and velocity estimation are taken into account". Where is this demonstrated?

This is indeed a hypothesis that currently cannot be demonstrated, as we have no data on speed and heading coding in this pathway. ‘We hypothesize that’ was added to the sentence to clarify this point (subsection “PG activity corresponds to observed path integration acuity”).

3) Title: The title is a bit misleading given that the egocentric-allocentric transformation is not explicitly demonstrated but rather suggested through simulation. Perhaps qualifying it with "A novel time-stamp mechanism could transform egocentric encounters into allocentric spatial representations".

We agree, and the title now reads: ‘A time-stamp mechanism may provide temporal information necessary for egocentric to allocentric spatial transformations’.

[Editors' note: further revisions were requested prior to acceptance, as described below.]

This paper contains the first recordings from neurons in weakly electric fish that project to the dorsolateral pallium (DL). DL is hypothesized to be the site of spatial memory, and this input is hypothesized to transform egocentric encounters into allocentric spatial representations.Essential revisions:1) There remains concern about how well the neural response model captures the measured neural responses. Figure 6—figure supplement 1 demonstrates that the predictions of the model and the measurements are positively correlated, and that this correlation is statistically significant; but this on its own is a relatively weak statement, and it is difficult to interpret the reported values of correlation coefficients. We would like to see a word of caution that the analysis relies on an assumption that the model correctly captures the neural responses.

A panel was added to Figure 6—figure supplement 1, namely panel A, to show examples of the measurements and the model’s prediction for an adapting and non-adapting cell.

In the panel depicting the distribution of correlation coefficients (now panel B), we added boxplots showing the control distribution for each cell (generated using random permutations).

Cautionary statement was added as suggested to the end of the Results section.

2) The mathematical analysis of temporal encoding precision assumes that β = 0, and accurate readout using the naive ML estimator requires approx. β < 0.2. Hence, it's relevant to know, what are the characteristic properties of cells with zero or small β: one can imagine a scenario in which these particular cells have very small gains, or short time constants, and are thus less useful for the computation than expected from panels A-C.

We added plots of the timescales, gains and baselines vs. *β* for all the cells as insets in Figure 6—figure supplement 1. Gain is negatively correlated with *β*, and hence memoryless cells have stronger responses which improves their temporal representation. Baseline activity c is not significantly correlated with *β*. The timescale is positively correlated with *β*, and so most cells with *β*<0.2 have 𝜏 in the range 1-10s. We added a cautionary note stating that we did not take into account the inter-relations between the model parameters (Results section).

3) The authors propose a hypothesis about how animals generate their representation of position relative to the environment, and could discuss how this prediction might be tested in future experiments, either behaviorally or in terms of neural activity in other brain areas. Behaviorally, the model implies that the animal's sense of position is critically dependent on the last encounter with an object. Does this make sense?

A paragraph was added to the Discussion section, discussing several possible lines of behavioral and physiological inquiry for future studies.

Another interesting prediction is that there may be a cutoff in the durations of swimming without encounters, over which the animal can estimate its position – determined by the distribution of adaptation time constants.

It's important to note that the range of timescales we report likely reflects our experimental paradigm, and not the actual distribution in the population. E.g., we report ~55% 'non-adaptive' cells, but this may very well include cell with 𝜏 <1s that our motor is too slow to probe. Likewise, cells with 𝜏 >30s simply stopped responding before we were able to record them. In other words, since our protocol tested intervals 1-30s long, it is no wonder this is the timescale range we encountered. To clarify this important point, a caveat regarding the time-scale distribution was added to the Results section.

[Editors' note: further revisions were requested prior to acceptance, as described below.]

We'd like to accept your paper, but first suggest you revise the paragraph excerpted below because it does not really address the reviewer's question.What we asked for in point 3 was "The authors propose a hypothesis about how animals generate their representation of position relative to the environment, and could discuss how this prediction might be tested in future experiments, either behaviorally or in terms of neural activity in other brain areas. Behaviorally, the model implies that the animal's sense of position is critically dependent on the last encounter with an object. Does this make sense?"You added a paragraph to the Discussion section, discussing several possible lines of behavioral and physiological inquiry for future studies. We would like a shorter reply that is more to the point in addressing this "Behaviorally, the model implies that the animal's sense of position is critically dependent on the last encounter with an object. Does this make sense?"

We replaced the last paragraph of the Discussion section with the text:

"In this contribution, we propose a hypothesis about how gymnotiform fish […] Combining these studies with chronic recordings of PG and its pallial targets in freely navigating fish will permit testing of our proposed space-to-time neural transformation scheme."

We kept the wording suggested by the reviewer(s) and editor. We agree that this is a concise summary of our core hypothesis as to how the fish 'knows where it is' with respect to environmental features. This suggested paragraph is much shorter than our previous suggestion, and we believe it is more to the point with regards to the reviewer's comments.